# Inhibition of adipocyte lipolysis by vaspin impairs thermoregulation in vivo

Inka Rapöhn [1,10], Helen Broghammer[1], Anne Hoffmann [1], Kevin Möhlis[1], Anna Moormann[1], Isabell Kaczmarek [2], Doreen Thor [2], Henning Großkopf [3], Laura Krieg[3], Isabel Karkossa [3], Kristin Schubert [3,4], Martin von Bergen [3,5], Kerstin Krause [6,7], Jana Breitfeld [6], Peter Kovacs [6], Nora Klöting[1], Rima Nuwayhid [8], Stefan Langer[8], Adhideb Ghosh[9], Christian Wolfrum [9], Michael Stumvoll [1,6], Matthias Blüher [1,6,7], John T. Heiker [1,5] ✉ & Juliane Weiner[6,10]

Altered activity of brown adipose tissue (BAT) contributes to obesity, insulin resistance, and cardiovascular disease. BAT secretes endocrine factors ("batokines") that regulate thermogenesis. We identify the serpin vaspin as a batokine that modulates adrenergic control of lipolysis and thermogenesis. Adipocyte-specific vaspin overexpression in mice reduces BAT activation and impairs thermoregulation during cold exposure or fasting. Mechanistically, vaspin binds low-density lipoprotein receptors (LRP1, LDLR, vLDLR), inhibiting adrenergic signaling and lipolysis in brown and white adipocytes by modulating phosphodiesterase activity and endocytic lipid uptake. Gene set enrichment analyses in human subcutaneous adipose tissue and in vitro studies confirm vaspin's anti-lipolytic effects in humans. Overall, vaspin emerges as a regulatory BATokine that fine-tunes BAT thermogenic activity to limit excessive energy expenditure and preserve metabolic balance.

Obesity and type 2 diabetes are becoming increasingly prevalent worldwide[1]. Accordingly, there is a great interest in a better understanding of the underlying mechanisms disrupting glucose and lipid metabolism to discover new potential treatment targets for these conditions. Reduced activity of brown adipose tissue (BAT) is associated with obesity, insulin resistance, and cardiovascular disease[2–4]. Furthermore, continuous sympathetic activation of white adipose tissue (WAT) leads to the formation of beige fat cells in rodents, a process known as browning or beiging of WAT[5,6]. These beige fat cells are the dominant BAT-like adipocytes in WAT of adult humans[7].

The main physiological function of brown and beige adipocytes is to control the amount of energy dissipated in form of heat that is required to maintain core body temperature, for example during cold exposure[8]. In brown adipocytes, this is achieved by directly uncoupling oxidative phosphorylation from ATP synthesis via the expression and activation of UCP1, while in beige adipocytes also futile cycling of creatine, free fatty acids or calcium drives non-shivering thermogenesis[9]. Targeting the activation of thermogenic adipocytes may thus enable to enhance energy expenditure via heat generation and could contribute to a negative energy balance helping to prevent

[1]Helmholtz Institute for Metabolic, Obesity and Vascular Research (HI-MAG) of the Helmholtz Zentrum München at the University of Leipzig and University Hospital Leipzig, Leipzig, Germany. [2]Rudolf Schönheimer Institute of Biochemistry, Medical Faculty, University of Leipzig, Leipzig, Germany. [3]Department Molecular Toxicology, Helmholtz Centre for Environmental Research (UFZ), Leipzig, Germany. [4]Cell Biology, Faculty of Environment and Natural Science, Brandenburg University of Technology Cottbus-Senftenberg, Senftenberg, Germany. [5]Institute of Biochemistry, Faculty of Life Sciences, University of Leipzig, Leipzig, Germany. [6]Leipzig University Medical Center, Divisions of Endocrinology and Nephrology, University of Leipzig, Leipzig, Germany. [7]German Center for Diabetes Research, Neuherberg, Germany. [8]Department of Orthopaedic, Trauma and Plastic Surgery, Division of Plastic, Aesthetic and Special Hand Surgery, University Hospital Leipzig, Leipzig, Germany. [9]Laboratory of Translational Nutrition Biology, Institute of Food, Nutrition and Health, ETH Zürich, Schwerzenbach, Switzerland. [10]These authors contributed equally: Inka Rapöhn, Juliane Weiner. ✉e-mail: john.heiker@helmholtz-munich.de

or reverse obesity. However, controlled heat production is essential, as uncontrolled thermogenic activation poses the risk of contributing to potentially life-threatening hyperthermia[10] or adipose and muscle wasting processes such as in cancer cachexia[11]. Consequently, understanding endogenous thermogenic brakes and developing pharmacological inhibitors for thermogenesis is equally significant for therapeutic purposes.

BAT releases specific endocrine signals, called brown adipokines or batokines, that may have tissue-specific functions[12,13]. Recent research has identified vaspin as part of the active brown adipocyte secretome in mouse and humans[14,15], suggesting a potential role in regulating adrenergic control of lipolysis and thermogenesis in vivo. Vaspin as an adipokine is primarily known for its anti-inflammatory role in WAT[16], protecting against adipocyte[17] and adipose tissue dysfunction and metabolic disease in diet-induced obese mice[18]. Its native serin protease inhibitor (serpin) function is thought to play a role in controlling protease activities, such as those of kallikrein 7 (KLK7)[19-22]. Vaspin also interacts with two cell surface molecules, mediating its functions independently of its serpin nature. In the liver, endothelial and kidney cells, vaspin interacts with the ER-resident heat shock protein GRP78, together with cell-specific coreceptors, to inhibit ER stress, fibrosis, and apoptosis under conditions of obesity and diabetes[23-25]. In white and brown adipocytes, vaspin binds to the LRP1 (low-density lipoprotein receptor-related protein 1) receptor[26]. This is followed by internalization, either resulting in lysosomal degradation or escape and translocation to the nucleus[27].

Here, we uncover the role of the vaspin-LRP1 system as an acute thermogenic brake, providing insights into the complex interplay between BAT, thermoregulation, and metabolic control.

## Results

### Vaspin transgenic mice exhibit impaired thermoregulation during acute cold exposure

We have previously shown that vaspin expression is increased in activated BAT in mice[14]. To elucidate the consequences of (brown) adipocyte vaspin expression on thermoregulation and BAT activity in vivo, we used a mouse model of adipose-specific vaspin overexpression (VasTg). VasTg mice express human vaspin in WAT and BAT, resulting in high vaspin serum levels (Fig. 1A). Transgenic vaspin expression is restricted to the adipocyte fraction of AT (Supplementary Fig. 1A), and no *Fabp4*-promoter leak expression was detected in other organs (Supplementary Fig. 1B). VasTg mice do not show significant differences in body weight and body composition compared to wild type (WT) littermates under standard conditions (chow diet and room temperature), as previously published[28]. However, oxygen consumption was lower over prolonged periods of time during light and dark phase and there was a trend for lower energy expenditure (p = 0.07 during active phase) in VasTg mice housed at room temperature (Supplementary Fig. 1C–D[28,]).

In this study, we challenged VasTg mice with conditions triggering adrenergic activation of lipolysis and thermogenesis. When exposed to cold, fed VasTg mice revealed significant cold sensitivity with lower body temperatures (~1 °C) during the first hours, before recovering (Fig. 1B). Acute cold exposure for 6 h in fasted mice reinforced the cold intolerance of VasTg mice with significantly lower body temperatures (~2 °C) and only partial recovery until the end of the experiment (Fig. 1C). Serum free fatty acid levels were significantly lower in VasTg mice (after 4 h cold exposure, Fig. 1D), while blood glucose levels were significantly higher after cold exposure, but not different between VasTg and WT mice (Fig. 1E). In BAT of cold exposed mice, we observed lower p38-MAPK and mTOR phosphorylation (Supplementary Fig. 1E–F). Expression of UCP1 and mitochondrial OXPHOS complexes (I, II, III and V) was not different (Supplementary Fig. 1E–H), as was expression of transgenic vaspin (Supplementary Fig. 1I). However, vaspin serum levels were significantly lower after acute cold exposure

but returned to thermoneutral levels after prolonged exposure (Fig. 1F). BAT histology revealed more lipid-laden brown adipocytes after longer cold-exposure in VasTg mice (Fig. 1G), that may be explained by impaired lipid mobilization. In line, expression of key thermogenic enzymes was significantly reduced (such as *Acly*, *Dio2*, or *Fasn*, but not *Ucp1*; Fig. 1H). Furthermore, muscle gene expression related to shivering thermogenesis (e.g. *Atp1a2*, *Gdp2*, *Ppard* and *Ryr1*) was significantly increased in cold-exposed VasTg (Fig. 1I), indicating compensation for impaired BAT thermogenesis. This was supported by microarray analysis, revealing extensively altered BAT gene expression (525 upregulated, 792 down regulated, p < 0.05; (Supplementary Fig. 1J and Supplementary Data 1), with lower expression of genes related to neuroactive ligand-receptor interactions, fatty acid biosynthesis, and insulin signaling (Fig. 1J and Supplementary Data 2). Genes with higher expression were enriched for the IL17 pathway (Fig. 1J and Supplementary Data 2), which could represent a compensatory mechanism of BAT activation via brown adipocyte IL17-RC mediated signaling[29]. Notably, *Rcan2* was by far the most differentially expressed gene (log2 FC = 4.6) and qPCR confirmed increased expression of *Rcan2* in BAT and WAT, but not liver or muscle of VasTg mice. However, *Rcan2* expression was unrelated to cold-exposure or affected by pharmacologic activation using CL (Supplementary Fig. 1K–L).

To exclude model specific effects, we performed acute injection of recombinant vaspin in WT C57BL/6 N mice. Intraperitoneal injection of recombinant vaspin (1 mg/kg) significantly increased serum vaspin levels up to 350 ng/ml 30 min post bolus (Supplementary Fig. 1M). Importantly and consistent with our findings in VasTg mice, vaspin-treated mice showed impaired thermoregulation during acute (fasted) cold-exposure with lower BAT and tail temperatures (Fig. 1K–N). In contrast to VasTg mice, vaspin-treated mice had significantly lower blood glucose levels after cold exposure, which may indicate an acute switch in substrate utilization towards glucose, compensating a potentially blunted lipolytic response (Fig. 1O). Furthermore, vaspin treatment did not alter *Rcan2* expression in BAT and WAT (Supplementary Fig. 1N), suggesting that the vaspin effect on BAT activation is independent of RCAN2. Rather, this seems to be a VasTg model-specific genetic alteration. Together, induced expression of vaspin in adipose tissue and injection of recombinant vaspin significantly blunted the thermogenic response in cold challenged mice.

### Vaspin induced changes in brown adipocyte signaling and proteome are linked to oxidative phosphorylation and thermogenesis

To gain broader insight into the cellular responses of brown adipocytes to vaspin, we analyzed the phosphoproteome and proteome of differentiated immortalized brown adipocytes (imBA) after treatment with recombinant vaspin (0.5 μg /ml) for 30 min (phosphoproteomics) and 6 h (proteomics) (Fig. 2A). We reliably quantified 2,203 unique phosphosites (in at least 3 biological replicates), and 393 (≈20%) were significantly regulated after 30 min of vaspin treatment (101 up and 292 down; Supplementary Fig. 2A and Supplementary Data 3). Proteins exhibiting lower phosphorylation states were significantly enriched for pathways related to thermogenesis and lipolysis (e.g. ADRB3, ADCY6, FABP4 and SERCA2), as well as mTOR and insulin signaling (KEGG mouse 2019, Fig. 2B and Supplementary Data 3-4). Analyzing the top regulated kinase-specific phosphorylation sites (20 of 97 total sites; Supplementary Fig. 2B and Supplementary Data 5), we found a higher phosphorylation state for kinases PKA and MYLK, whereas phosphorylation states of KSR1, mitogen activated kinase Map3k3, CAMK2D, the cyclin-dependent kinase CDK7 (lower phosphorylation state) were lower. Phosphorylation of PKA occurred at serine 83 of the regulatory subunit and this has been shown to inhibit PKA activity[30]. On the proteome level, we reliably quantified 2,327 proteins in total, and 200 (112 higher / 88 lower) after 6 h with significantly altered abundances in

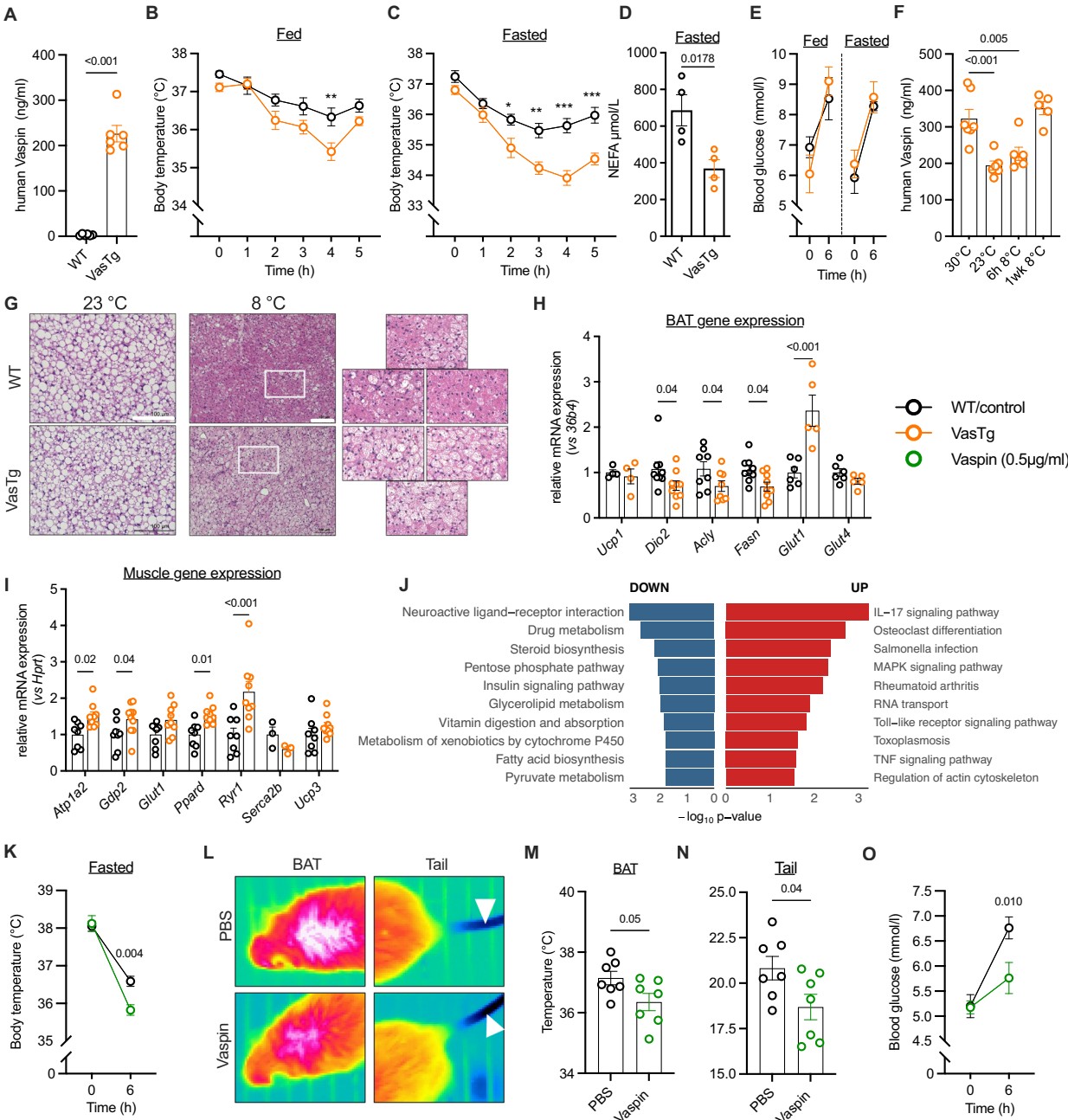

**Fig. 1 | VasTg mice show impaired thermoregulation. A** Serum levels of human vaspin (SERPINA12) measured by ELISA of VasTg and WT mice ($n$ = 6/6). **B, C** Rectal body temperature during acute cold exposure in fed (**B**), $n$ = 9/9) and fasted (**C**), $n$ = 8/7) VasTg and WT mice. **D** Serum NEFA levels of VasTg and WT mice after acute (4 h) cold exposure ($n$ = 4/4). **E** Blood glucose levels of VasTg and WT mice before and after acute (6 h) cold exposure ($n$ = 4/4). **F** Serum levels of human vaspin (SERPINA12) of VasTg housed at 30 °C, 23 °C, for 6 h at 8 °C and for one week at 8 °C measured by ELISA ($n$ = 7/6/6/5). **G** Representative H&E-stained sections of BAT (20x; and magnifications) from VasTg and WT mice housed at 23 °C and 8 °C for 7 days (close ups from indicated area and 2 additional animals per group). **H** mRNA expression of thermogenic and metabolic genes in BAT of VasTg and WT mice after cold exposure (24 h; $n$ = 4–10). Gene expression is relative to controls, normalized to *36b4*. **I** mRNA expression of genes related to shivering thermogenesis in muscle of VasTg and WT mice after cold exposure (24 h; $n$ = 9 per group). Gene expression is relative to controls, normalized to *Hprt*. **J** KEGG pathway functional enrichment analysis of DEGs. The vertical axes represent the KEGG pathways significantly enriched; the horizontal axis indicates -log10(p-value). **K–N** Rectal body temperature (**K**), thermal images from BAT and tail (**L**), BAT surface (**M**) and tail surface (**N**) temperatures, as well as blood glucose levels (**O**) in fasted vaspin-treated (intraperitoneally, 1 mg/kg) and control C57BL/6 N mice ($n$ = 7/8 per group) before and after acute (6 h) cold exposure. Data are shown as mean ± SEM. WT or control samples are indicated as black circles, VasTg samples as orange circles and Vaspin-treated samples as green circles. Statistical significance was evaluated by two-way ANOVA with Šídák's (**B, C, J, K, O**) or Dunnett's (**F**) post-hoc test or uncorrected Fischer's LSD (**H, I**), or unpaired two-sided t-tests (**A, D, M, N**). *$p$ value < 0.05, **$p$ value < 0.01, ***$p$ value < 0.001.

vaspin treated adipocytes (Supplementary Fig. 2A and Supplementary Data 6). Mitochondrial proteins were significantly enriched among all regulated proteins (GO Cellular Compartment, Supplementary Fig. 2C). Among the proteins with lower abundance, pathway analysis

revealed significant enrichment of proteins related to oxidative phosphorylation, such as core subunits of respiratory chain complex I (NDUFS6, NDUFA3, NDUFA12 and NDUFB8), complex III (UQCRFS1), and COX7A2, the terminal oxidase in mitochondrial electron transport

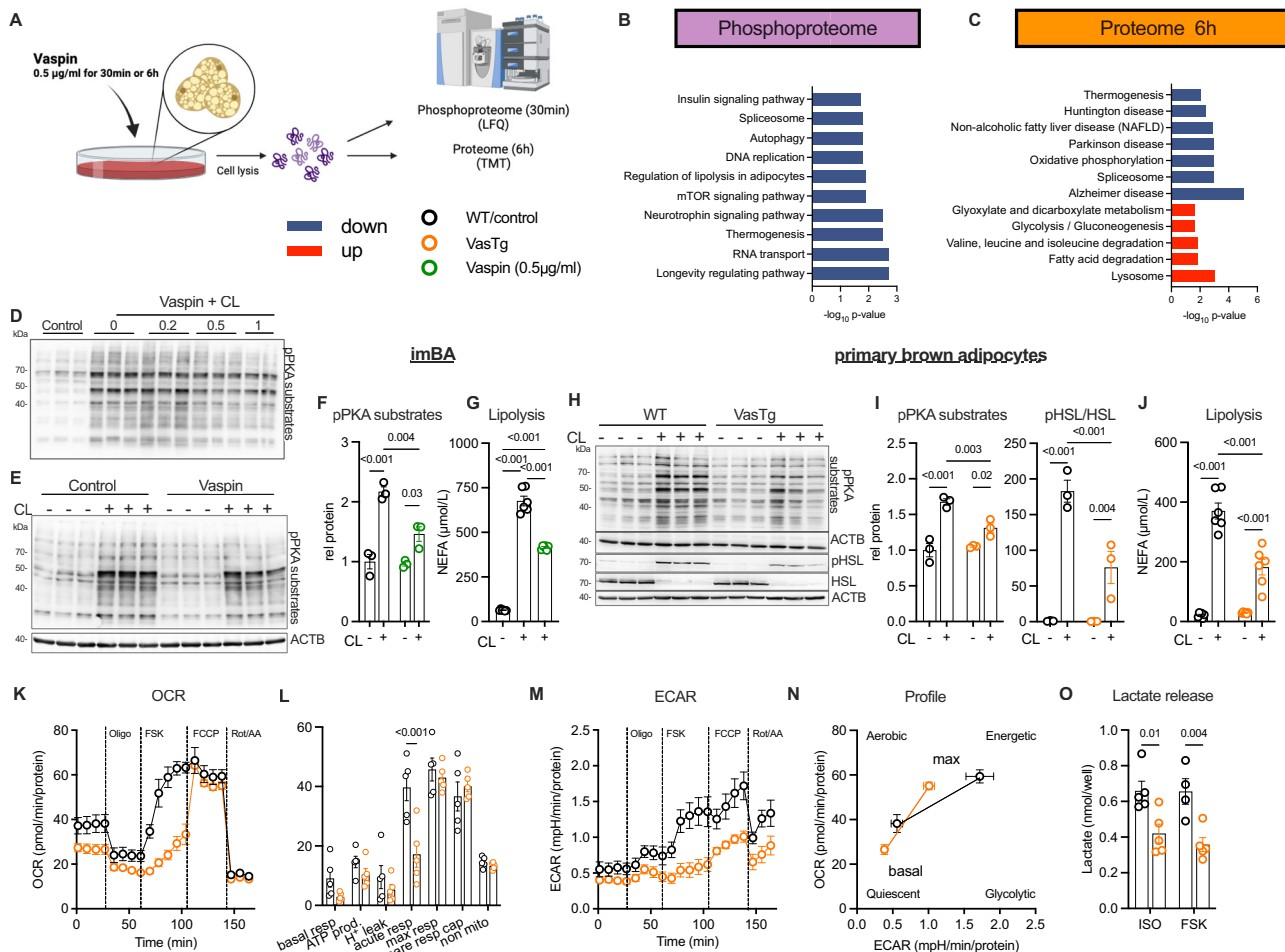

**Fig. 2 | Vaspin induces changes in cellular (phospho-) proteome and inhibits activation of brown adipocyte metabolism and thermogenesis. A** Experimental design (created in BioRender, https://BioRender.com/sxodmnd). Differentiated immortalized brown adipocytes (imBA) were exposed to recombinant vaspin (0.5 μg/mL) for 30 min or 6 h, respectively. Proteome and phosphoproteome were created from proteins of the same samples. **B, C** KEGG pathway functional enrichment analyses of affected phosphorylation sites (after 30 min, **B** and protein abundances (after 6 h, (**C**). The vertical axis represents KEGG pathways significantly enriched, and the horizontal axis indicates -log10 (p-value). Significantly enriched KEGG pathways are labeled in red (increased) and blue (decreased). For details, please see "Methods" section. **D** Western blot analysis of basal and CL-induced PKA-activation in vaspin treated (0.2 – 1 μg/ml) imBA cells. **E–G** Western blot analysis and (**F**) quantification of basal and CL-induced PKA-activation as well as (**G**) free fatty acid release (NEFA, **G**), n = 6 per condition) in vaspin treated differentiated imBA. **H–J** Western blot analysis and (**I**) quantification of basal and CL-induced PKA and

HSL activation as well as (**J**) free fatty acid release (NEFA), n = 6 per condition) in differentiated primary brown adipocytes from VasTg and WT mice. **K, L** Time-resolved OCR (**K**) of differentiated primary brown adipocytes from VasTg and WT mice measured by Seahorse (representative experiment, n = 5/5) and (**L**) quantification of basal respiration, ATP production, proton leak, acute response to FSK, maximum and spare respiratory capacity and non-mitochondrial respiration. **M** Time-resolved ECAR corresponding to the OCR shown in (**K**). **N** Energetic profile (OCR / ECAR plot from K and M) of differentiated primary brown adipocytes from VasTg and WT mice. **O** Lactate release from ISO and FSK treated primary brown adipocytes from VasTg and WT mice (n = 5/5). WT or control samples are indicated as black circles, VasTg samples as orange circles and Vaspin-treated samples as green circles. Data are presented as mean ± SEM of at least two (**D, G**) or three (**E, F, H–N**) independent experiments. Statistical significance was evaluated by two-way ANOVA with Šídák's (**L, O**) or Tukey's (**F, G, J**) post-hoc test or uncorrected Fischer's LSD (I). *p value < 0.05, **p value < 0.01, ***p value < 0.001.

(KEGG mouse 2019, Fig. 2C and Supplementary Data 7). Whole BAT from VasTg mice had not shown significant alterations in OXPHOS protein expression (Supplementary Fig. 1E–H), however analyses of bulk tissue may obscure adipocyte-specific changes due to contributions from other cell types and the OXPHOS antibody cocktail lacks resolution needed to validate proteomic findings of changes in specific OXPHOS proteins and subunits.

Upregulated proteins were related to fatty acid metabolism (FASN, ACADSB and CRAT), acetyl-CoA acyltransferases (ACAT1 and ACAA2) and branched chain amino acid metabolism (ALDH7A1 and GLUL), suggesting compensatory adaptations for energy substrate supply. In addition, lysosomal proteins were upregulated, supporting previously reported vaspin internalization and degradation in adipocytes (Fig. 2C)[26]. Taken together, vaspin treatment significantly impacted intracellular signaling pathways, leading to proteomic

changes in brown adipocytes, that likely affect brown adipocyte metabolism and function.

## Vaspin inhibits adrenergic control of lipolysis and mitochondrial respiration in brown adipocytes

We then investigated the acute effects of vaspin on brown adipocyte activation in vitro, first analyzing immortalized brown adipocytes (imBA) in the basal and ADRB3-stimulated state (using CL316,243) after treatment with recombinant vaspin (0.2–1 μg/mL). We found that vaspin dose-dependently suppressed ADRB3-mediated intracellular PKA activity (Fig. 2D–F) and lipolysis (Fig. 2G, using 0.5 μg/ml in this and all subsequent experiments). Also, in vaspin-treated mice, PKA-substrate phosphorylation in BAT was significantly lower compared to controls (Supplementary Fig. 2D–E), without significant differences in BAT expression of *Ucp1* and core mitochondrial proteins

(Supplementary Fig. 2F–H). These findings were furthermore confirmed in primary brown adipocytes from both WT (Supplementary Fig. 2I–K) and *Ucp1*-knockout mice (Supplementary Fig. 2L–P), precluding the dependence on UCP1 expression.

Next, we were interested whether vaspin may function in an autocrine/paracrine manner, therefore using primary brown adipocytes from VasTg mice and littermate controls[28]. VasTg adipocytes endogenously express human vaspin and secreted vaspin up to levels of 80 ng/ml under experimental conditions (Supplementary Fig. 3A). VasTg primary brown adipocytes differentiated like control cells (Supplementary Fig. 3B-C) yet exhibited lower expression of thermogenic *Adrb3* and *Ucp1* than control adipocytes (Supplementary Fig. 3D). As observed for exogenous vaspin in imBA, endogenously vaspin expressing VasTg adipocytes showed a blunted response to ADRB3-activation, with lower PKA and p38MAPK activation, and downstream lipolysis (Fig. 2H-J). To exclude potential effects of reduced *Adrb3* expression, we repeated experiments stimulating VasTg primary brown adipocytes with norepinephrine (NE; activating all adrenergic receptors, ADRA and ADRB), or isoproterenol (ISO; only activating ADRBs), and forskolin (FSK; directly activating adenylate cyclases). Under these conditions, vaspin expression inhibited activation of PKA, downstream phosphorylation of HSL and p38MAPK, and cellular lipolysis and NEFA release (Supplementary Fig. 3E-J). Together, vaspin significantly inhibits cAMP-dependent signaling, whether exogenously added or endogenously expressed and secreted by brown adipocytes.

We then assessed the impact of vaspin on brown adipocyte mitochondrial respiration, including the acute response to FSK, in VasTg brown adipocytes using a Seahorse analyzer (Fig. 2K). While there were no significant differences in basal and maximal respiration between brown adipocytes from VasTg and WT mice, the acute response to FSK was significantly blunted (Fig. 2L). There were no differences in mitochondrial OXPHOS protein content (Supplementary Fig. 3K–L). Extracellular acidification rates (ECAR) were significantly lower, especially under stressed conditions (FSK and FCCP; Fig. 2M). Plotting oxygen consumption rates (OCR) vs. ECAR under basal and maximal respiration, revealed a more quiescent energetic profile of VasTg-derived brown adipocytes in the basal state and a significantly lower glycolytic phenotype under stressed conditions (Fig. 2N). Consistently, lactate release was significantly lower in ISO and FSK treated VasTg primary brown adipocytes (Fig. 2O), suggesting an attenuated glycolytic flux.

## Vaspin inhibition of adrenergic signaling involves binding LDL receptors and PDE activities

On the cellular level, regulation of the adrenergic signaling axis involves multiple key molecules. To address these cellular nodes more precisely, we stimulated primary and immortalized brown adipocytes using specific activators or inhibitors of distinct pathways (Fig. 3A). First, we investigated whether secretion of vaspin was necessary or if vaspin may also act intracellularly to inhibit acute lipolytic activity. We therefore treated primary brown adipocytes of VasTg and control littermates with brefeldin A (BFA) to suppress secretion prior to CL treatment (Fig. 3B). In line with an extracellular function of vaspin, BFA treatment abolished the inhibitory effect of vaspin expression in VasTg brown adipocytes on PKA activation and lipolysis (Fig. 3C-E).

Given the acute effects of extracellular vaspin on adipocyte function, we speculated on involvement of a cell surface receptor transducing the extracellular signal into the adipocytes. We first focused on the LRP1, that was recently identified as the receptor mediating rapid endocytosis of vaspin in adipocytes[26]. As previously shown, BAT *Lrp1* expression is significantly reduced in cold-exposed WT mice (Fig. 3F and ref. [31]). We established siRNA-mediated knock down of *Lrp1* expression in differentiated brown adipocytes, achieving >80% reduction in protein expression (Fig. 3G–H) and lack of LRP1

abolished vaspin inhibition of CL-induced PKA activation and lipolysis (Fig. 3G, I, J). Vaspin binding to the LRP1 requires initial contact to cell-surface glycosaminoglycans and a non-heparin-binding vaspin variant does not internalize in LRP1 expressing adipocytes[26]. Consistent with a mechanism involving LRP1 binding, the non-heparin-binding (NHB) vaspin mutant had no effect on CL-induced PKA activation in WT primary brown adipocytes (Fig. 3K–L). To disseminate whether this inhibitory effect on adrenergic signaling and lipolysis mediated by LRP1 is specific to vaspin, we repeated experiments using receptor-associated protein (RAP). RAP is the endogenous ligand of LRP1, preventing ligand interactions of LDL receptors during cellular trafficking. LRP1-bound RAP is rapidly internalized[26], but had no effect on the adrenergic response of brown adipocytes to CL (Fig. 3M). Finally, pretreating imBAs with RAP to reduce LRP1 receptor density on the cell surface or with chlorpromazine (CPZ) to inhibit receptor internalization, both abolished vaspin inhibition of CL-stimulated lipolysis (Fig. 3N). These findings suggest that vaspin needs to interact with the LRP1 and that subsequent internalization and signaling events inhibit the lipolytic response of brown adipocytes.

LDL receptor family members, including LRP1, have been shown to interact and regulate GPCR signaling, such as Gai-coupled CXCR4, both activating and inhibiting ligand induced signaling[32]. We investigated whether the vaspin effect was depending on Gai-signaling (whether via ADRAs or other Gai-coupled GPCR. However, we found that pertussis toxin (PTX) did not affect vaspin inhibition of NE-induced PKA activation (Fig. 3O–P). We next assessed whether vaspin may modulate activities of phosphodiesterases (PDE). Indeed, in the presence of PDE inhibitor IBMX in imBA cells, the inhibitory effect of vaspin on adrenergic activation of PKA was abrogated (Fig. 3Q–R). We then used specific inhibitors for the two major adipocyte PDEs in Ro 20-1724 (Ro, PDE4) and Cilostamid (Cilo, PDE3) to address potential involvement of these PDEs. Interestingly, inhibition of PDE3 (by Cilo) did not affect vaspins' suppression of CL-induced PKA activation, but inhibition of PDE4 (by Ro) did rescue CL effects in VasTg primary brown adipocytes (Fig. 3S, T). In differentiated imBA, all PDE inhibitors increased CL-induced lipolysis and free fatty acid release (Fig. 3U), and consistently vaspin inhibition of lipolysis was only abolished by IBMX and Ro, and not Cilo (Fig. 3U).

Finally, thermogenic adipocytes also rely on lysosomal lipolysis from autophagocytozed lipid droplets or endocytosed lipoproteins. Lipoprotein uptake is mediated by members of the LDL receptor, mainly LDLR, vLDLR and LRP1[33,34]. Intrigued by vaspin's high affinity binding to LRP1, we investigated whether vaspin may bind other members of the LDL receptors family as well. Indeed, using a previously established ELISA-based binding assay, we found low or sub nanomolar binding to core members LDLR (EC$_{50}$ = 43 nM, 95%CI = 32-58 nM) and vLDLR (EC$_{50}$ = 2 nM, 95%CI = 0.8–4.5 nM, Fig. 3V, Table 1), and to LRP2 and LRP4 (EC$_{50}$ = 0.6 nM, 95%CI = 0.1 -2.2 nM and EC$_{50}$ = 0.4 nM, 95%CI = 0.1 – 1.1 nM, respectively; Fig. 3W, Table 1). Vaspin did not bind distant and far distant members LRP5, LRP6, and LRP10, as well as class B scavenger receptors CD36 and SR-BI (Fig. 3X, Table 1).

Taken together, these data suggest that vaspin exerts multiple mechanisms to control adrenergic activation of brown adipocyte metabolism and lipid turnover through interactions with the LDLR family, thereby inhibiting cAMP-dependent neutral lipolysis via modulation of PDE activities and blocking endocytotic lipid trafficking towards lysosomal acid lipolysis.

## Vaspin suppresses activation of lipolysis in mouse and human WAT

Sympathetic tone in WAT is increased e.g. during short-term fasting to induce WAT lipolysis in combination with lower fasting insulin levels. The WAT-derived energy substrate supply in form of free fatty acids has been shown to be indispensable for BAT thermogenesis and

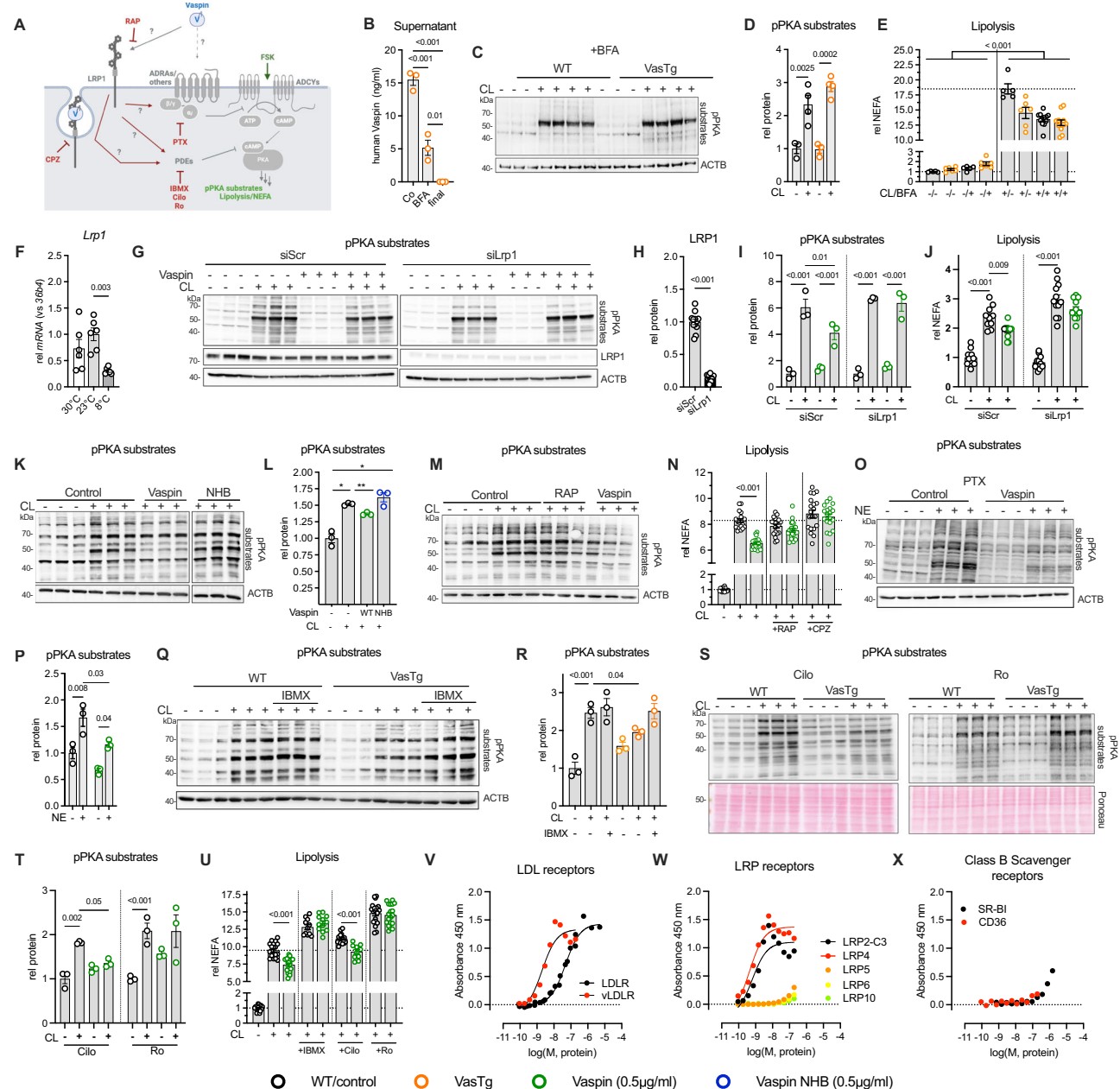

thermoregulation during acute fasting[35]. To investigate whether adipocyte-specific vaspin expression affects WAT lipid mobilization during fasting, we challenged VasTg mice with a fast-refeed experiment. Notably, after the fasting period VasTg mice had significantly lower body temperature compared to WT mice (Fig. 4A), with significantly lower BAT and tail temperatures (Fig. 4B–D). Also, blood glucose levels were significantly lower in VasTg mice (Fig. 4E). After refeeding, VasTg mice recovered to pre-fast levels without differences to WT controls. These data suggest that vaspin's suppressive effect on adrenergically stimulated lipolysis is not restricted to brown adipocytes. We thus investigated vaspin effects on lipolysis in mouse primary white adipocytes from iWAT and eWAT and found a blunted response to ADRB3-stimulation with lower PKA activation (Fig. 4F–I). To assess whether vaspin (*SERPINA12*) expression may be related to pathways of lipid mobilization and energy expenditure, we performed gene set enrichment analyses in human subcutaneous adipose tissue (SAT) of 1143 patients from the LOBB. These indeed revealed significant negative correlations of *SERPINA12*/vaspin expression with genes related to the TCA cycle and oxidative

phosphorylation (Fig. 4J). We confirmed the molecular interaction in human cells using fluorescently-labeled vaspin that colocalized with LRP1 in mature human SAT adipocytes (Fig. 4K) and established differentiation of human SVF-derived adipocytes (Fig. 4L). While, and not unexpected, there was considerable heterogeneity in basal lipolysis and adrenergic responsiveness in SAT SVF-derived adipocytes from individual patients (Fig. 4M), vaspin significantly inhibited ISO-induced activation of PKA and lipolysis in SAT SVF-derived adipocytes from two patients (male and female), whose adipocytes exhibited normal basal lipolytic activity and sensitivity to ISO, and that originated from distinct AT locations (Fig. 4M–O, and Supplementary Table 1). It did not exhibit an effect in adipocytes from patients with adipocytes that showed resistance to ISO-stimulation (hA1-0039) or that exhibited high basal lipolytic activity (hA1-0037) (Fig. 4M). Taken together, our data support acute and suppressive effects of vaspin on BAT and WAT lipolysis, resulting in limited BAT activity and impaired thermoregulation under conditions requiring rapid supply of energy substrates, such as cold exposure or fasting. Mechanistically, inhibition of adrenergic activation of WAT and BAT

**Fig. 3 | Vaspin inhibition of adrenergic signaling involves LRP1 binding and modulation of PDE activities. A** Regulatory mechanisms of adrenergic signaling and levels of intervention (created in BioRender, https://BioRender.com/sxodmnd). **B** Cell culture supernatant of human vaspin (SERPINA12) after 3 h starvation (control) and parallel blocking of vaspin secretion using BFA (BFA–after 3 h prior medium change, final–post stimulation with CL) before signal transduction and lipolysis assays in differentiated primary brown adipocytes from VasTg measured by ELISA ($n = 3$ per condition). **C–E** Western blot analysis and (**D**) quantification of basal and CL-induced PKA-activation ($n = 3$ (basal) and 4 (CL)) as well as (**E**) free fatty acid release in differentiated primary brown adipocytes from VasTg and WT mice after blocking vaspin secretion (NEFA, $n = 5$–12 per condition). **F** BAT *Lrp1* expression in WT mice housed at 30 °C, 23 °C or 8 °C ($n = 6$ per temperature). **G–J** Knockdown of LRP1: (**G**) Western blot analysis and (**H**) quantification of LRP1 expression, (**I**) basal and CL-induced PKA-activation as well as (**J**) free fatty acid release (NEFA, $n = 10$–12 per condition, please see source data file) in vaspin-treated differentiated imBA, with or without siRNA-mediated *Lrp1* knockdown. **K, L** Non-heparin-binding (NHB) vaspin variant: (**K**) Western blot analysis and (**L**) quantification basal and CL-induced PKA-activation in vaspin and NHB-treated and control differentiated imBA ($n = 3/3$). **M** LRP1 ligand RAP: Western blot analysis of basal and CL-induced PKA-activation in RAP or vaspin treated differentiated imBA. **N** Blocking of vaspin binding to LRP1 (RAP preincubation) or LRP1 internalization by clathrin-mediated endocytosis (using CPZ): Quantification of basal and CL-induced free fatty acid release with or without vaspin treatment in differentiated imBA ($n = 19$–20 per condition, please see source data file). **O, P** Inhibition of Gas

signaling using PTX: (**O**) Western blot analysis and (**P**) quantification of basal and NE-induced PKA-activation in vaspin-treated differentiated imBA with PTX pre-treatment ($n = 3/3$). **Q, R** Inhibition of PDE activities using IMBX: (**Q**) Western blot analysis and (**R**) quantification of basal and CL-induced PKA-activation in differentiated primary brown adipocytes from VasTg and WT mice ($n = 3/3$). **S–U** Inhibition of PDE3 and PDE4 using Cilo and Ro: S) Western blot analysis and (**T**) quantification of basal and CL-induced PKA-activation in differentiated primary brown adipocytes from VasTg and WT mice with Cilo or Ro pretreatment and controls ($n = 3/3/3$). **U** Inhibition of total PDE activities using IBMX, Cilo or Ro: Quantification of basal and CL-induced free fatty acid release with or without vaspin treatment in differentiated imBA ($n = 10$–20 per condition, please see source data file). **V-X** ELISA-based analysis of TAMRA-vaspin binding to LDL receptors (**V**) - LDLR in black, vLDLR in red; LRP receptors (**W**) - LRP2 in black, LRP4 in red, LRP5 in orange, LRP6 in yellow, LRP10 in green); and class B scavenger receptors (**X**) - SR-BI in black, CD36 in red. Nonlinear regression analysis was performed to determine $EC_{50}$ presented in Table 1. Data are presented as mean ± SEM of at least two (**B–E, J–L, M–U**) or three (**G–I**) independent experiments. WT or control samples are indicated as black circles, VasTg samples as orange circles, Vaspin-treated samples as green circles, Vaspin NHB-treated samples as blue circles. Statistical significance was evaluated by one-way ANOVA with Šídák's (**B, D, E, L**) or uncorrected Fischer's LSD (**N, R, U**), or two-way ANOVA with Dunett's (**I**) or Tukey's (**F, J, P**) post-hoc or uncorrected Fischer's LSD (**T**), or unpaired two-sided t-test (**H**). *$p$ value < 0.05, **$p$ value < 0.01, ***$p$ value < 0.001.

**Table 1 | $EC_{50}$ values (nM) for the interaction of vaspin with members of the LDL receptor family and Class B scavenger receptor**

|          | Vaspin          | n |
|----------|-----------------|---|
| LDLR     | 43.3 (32.2–57.8)| 3 |
| vLDLR    | 2.0 (0.8–4.5)   | 1 |
| LRP2-C3  | 0.6 (0.1–2.2)   | 1 |
| LRP4     | 0.4 (0.1–1.1)   | 1 |
| LRP5     | n.b.            | 1 |
| LRP6     | n.b.            | 1 |
| LRP10    | n.b.            | 1 |
| CD36     | n.b.            | 1 |
| SB-RI    | n.b.            | 1 |

All $EC_{50}$ (nM) data are presented with 95% confidence interval obtained from n = 1-3 independent experiments with 4 technical replicates. n.b. = no binding.

lipolysis limits fuel supply for BAT mitochondrial respiration, UCP1 activation and thermogenesis.

## Discussion

We previously identified vaspin as a serpin that is increasingly expressed and secreted during brown adipocyte adipogenesis and after BAT activation in response to cold exposure or feeding on a high-fat or high-sucrose diet[14,36]. Gene expression analysis also revealed that vaspin is specifically increased in beige WAT (after cold exposure), suggesting its functional relevance in thermogenic adipocytes in general[36]. Studies on vaspin in white AT have established its protective role in counteracting obesity-induced inflammation and insulin resistance[17,19,25], at least in part via the inhibition of pro-inflammatory proteases such as KLK7[22]. However, there has been no evidence of a functional role for vaspin in the BAT-mediated control of adaptive energy expenditure. Here, we demonstrate that vaspin is involved in the adrenergic control of BAT activation and inhibits thermogenesis in vivo.

Our experimental data in cell cultures demonstrate local, auto- or paracrine, and receptor-mediated functions of vaspin in activated BAT. Mechanistically, vaspin appears to inhibit the adrenergic control of BAT activity by mechanisms involving PDEs, abrogating the cAMP signal, and subsequently lowering activities of kinases such as PKA and

p38MAPK, that are key drivers of BAT metabolism, transcription and thermogenic potential (Fig. 5). Changes in the brown adipocyte phosphoproteome induced by vaspin revealed increased phosphorylation of PKA at the inhibitory site, Ser83[30]. ERK2 controls lipolysis by phosphorylating ADRB3 (at Ser247) and potentially other proteins, that regulate cAMP levels, such as ADCY6 and PDE4A/B[37]. In brown adipocytes, ADRB3-mediated ERK1/2 activation depends entirely on the cAMP-PKA axis[38], and consistent with this, we observed significantly lower levels of phosphorylated ADRB3 (Ser247) and ADCY6 (Ser574) in vaspin-treated brown adipocytes (Supplementary Data 3).

The rapid alterations in intracellular phosphorylation states induced by vaspin also suggest the involvement of a cell surface receptor. In this regard, we have recently identified LRP1 as the primary endocytosis receptor for vaspin in white and brown adipocytes[26]. Here, we demonstrate that vaspin acts as an acute thermogenic brake, a mechanism that appears to be independent of its serpin function and instead relies on the interaction with the LRP1 receptor. Significantly lower blood vaspin levels following cold exposure[14], lower levels of vaspin in the supernatant of NE-stimulated human brown adipocytes[15], and the inverse regulation of thermogenic activation and expression of the LRP1 receptor in brown adipocytes[31] further support this mechanism. Vaspin inhibition of adrenergic signaling required LRP1 binding and was independent of intracellular inhibitory G protein signaling but was suppressed by inhibition of brown adipocyte PDEs using IBMX. The highest expression in AT has been demonstrated for PDE3 and PDE4 and, specifically in BAT, PDE4 has been shown to primarily control adrenergic stimulation of lipolysis, while PDE3 predominantly regulates the induction of *Ucp1* expression[39]. Consistently, vaspin inhibition of adrenergically induced lipolysis depended on PDE4 activity rather than PDE3. Notably, the expression of *Pde4b* and PDE4-anchoring protein myomegalin (*Pde4dip*) was significantly higher in VasTg mice (Supplementary Data 1). We hypothesize that, via the LRP1, vaspin may affect cAMP levels, substantially dampening the BAT thermogenic response. The interaction of LRP1 with MAPK kinases ERK1 and ERK2[40,41] may provide a link to the regulation of PDE4; for example, ERK2-mediated phosphorylation of PDE4B and PDE4D isoenzymes can activate or inhibit PDE4 activity[42,43]. Related to this, lactoferrin was shown to induce white adipocyte lipolysis by binding to the LRP1 and activating ERK signaling to increase cAMP levels either involving or not involving Gas signaling, depending on the cell type investigated[44,45]. The dynamics of LRP1 distribution within the cell-

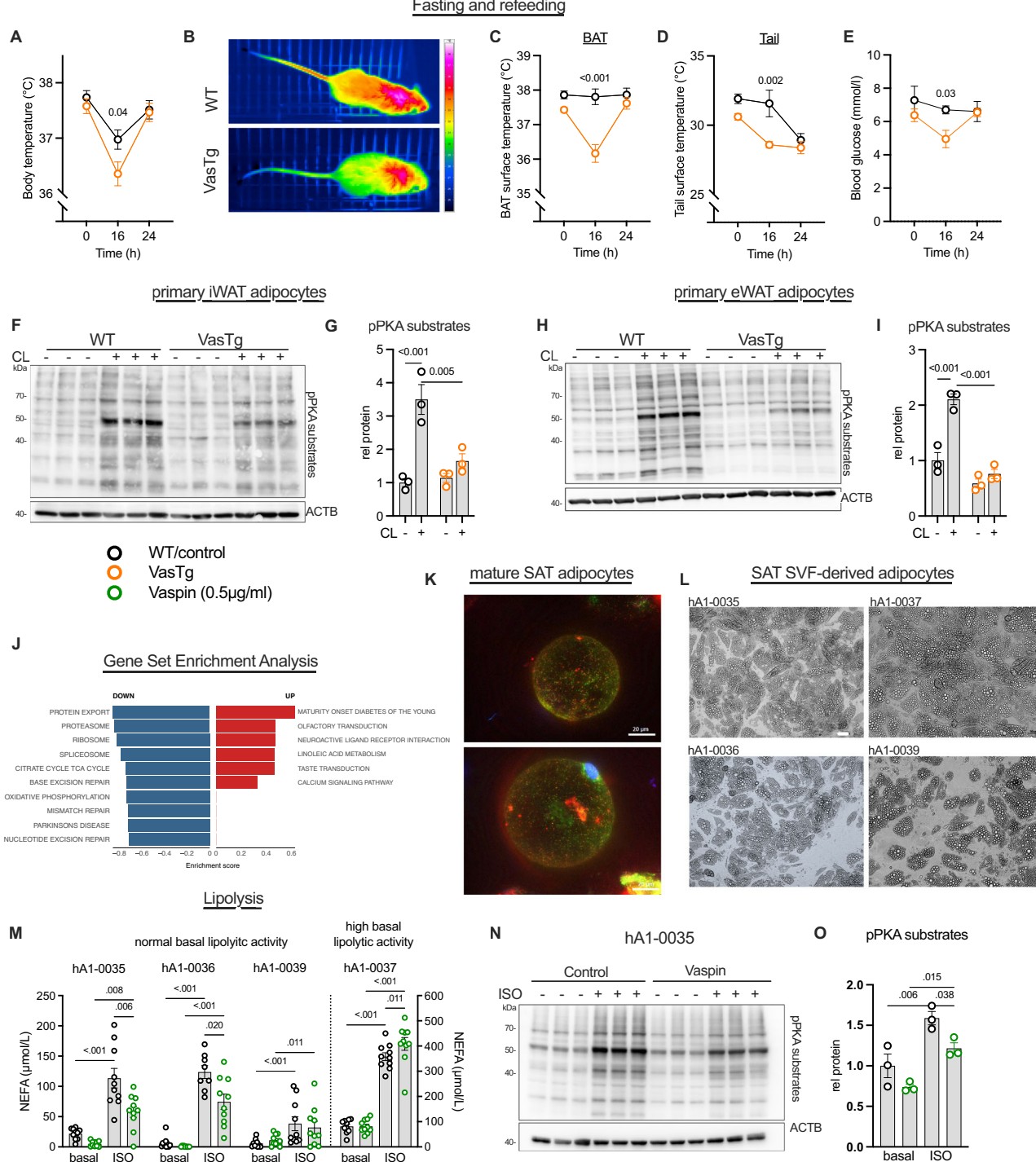

**Fig. 4 | Vaspin suppresses adrenergic activation and lipolysis of white adipose tissue in mice and humans. A–E** Rectal body temperature (**A**), thermal images from BAT (**B**), BAT surface (**C**) and tail surface (**D**) temperatures, as well as blood glucose levels (**E**) in VasTg and WT during a fasting-refeeding cycle (**A**): $n = 8/9$, **C**, **D**: $n = 4/4$). **F–I** Western blot analysis (**F**, **H**) and quantification (**G**, **I**) of basal and CL-induced PKA-activation in differentiated primary iWAT (**F**, **G**) and eWAT (**H**, **I**) adipocytes from VasTg and WT mice. **J** Gene set enrichment analysis (GSEA, using KEGG pathways) of genes correlating with *SERPINA12* in subcutaneous adipose tissue (SAT). The top 10 enriched pathways for positively and negatively corre-lating genes are shown sorted by adj. p and set size (please see "Methods" section). **K** Overlay images of immunofluorescence microscopy images of human mature SAT adipocytes from one patient stained for LRP1 (green) and treated with TAMRA-labeled vaspin (red). **L** Brightfield images of differentiated SAT SVF-derived human adipocytes from four different patients. **M** Basal and ISO-induced free fatty acid release in vaspin treated (0.5 μg/ml) differentiated SAT SVF-derived human adi-pocytes from four patients ($n = 9$-$10$ per condition and patient, please see source data file). **N**, **O** Western blot analysis (**I**) and quantification (**J**) of basal and ISO-induced PKA-activation of differentiated SAT SVF-derived human adipocytes. Data are presented as mean ± SEM of at least two (independent experiments (**G**, **I**) or from one the four patients analyzed (**O**). Statistical significance was evaluated by two-way ANOVA with Šídák's (**A–D**) or Tukey's (**G**, **I**) post-hoc test or uncorrected Fischer's LSD (**E**, **M**), or one-way ANOVA with uncorrected Fischer's LSD (**O**). *$p$ value < 0.05, **$p$ value < 0.01, ***$p$ value < 0.001. Scale bar: 50 μm.

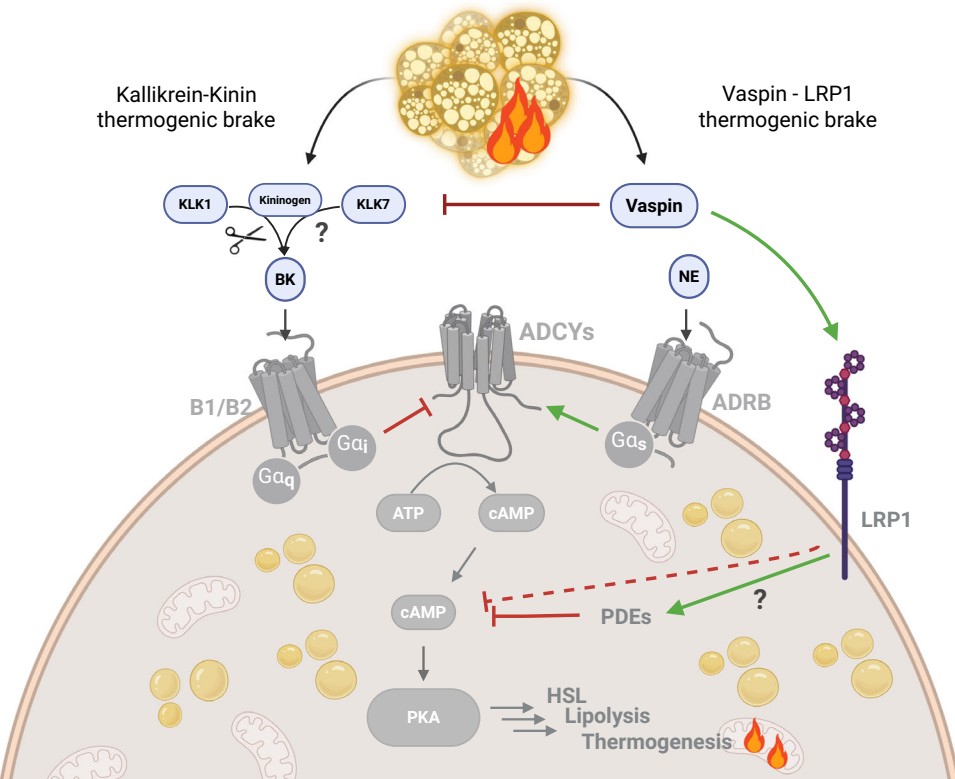

**Fig. 5 | Proposed mechanisms and interplay of thermogenic brakes via the vaspin-LRP1 and kallikrein-kinin pathways.** The vaspin-LRP1 thermogenic brake is reported here, together with possible cross talk with the kallikrein-kinin thermogenic brake reported by Peyrou and colleagues [53]. *ADCY* adenylate cyclase, *ADRB* adrenergic receptor, *BK* bradikinin, *B1/B2* bradykinin receptors B1/B2, *Ga* G-alpha protein, *KLK* kallikrein, *LRP1* low-density lipoprotein receptor 1, *NE* nor-epinephrine, *PDE* phosphodiesterase, *PKA* protein kinase A. Created in BioRender, https://BioRender.com/sxodmnd.

membrane, whether localized in lipid rafts or lipid raft-free membrane compartments, are believed to be key to the receptor's numerous ligand- and tissue/cell-specific signaling activities (lipid rafts) or endocytotic action (raft-free areas), and their regulation[46]. Interestingly, inhibiting clathrin-dependent endocytosis abrogated the inhibitory effect of vaspin on lipolysis. This suggests that co-internalization of entire membrane protein complexes or intracellular protein scaffolds that interact with the LRP1 may be required for the vaspin effect in thermogenic adipocytes[47]. Protein complexes, co-internalized with the respective scaffold receptor, remain as active signaling complexes on the surface of endosomal vesicles[48,49], which are essential for regulating localized signaling, as demonstrated for MAPK/ERK signaling[50,51]. Clearly, further research is needed to understand these distinct ligand- and cell-specific signaling events initiated by LRP1 binding.

In addition to neutral lipolysis, thermogenic adipocytes also rely on lysosomal acid lipolysis of autophagocytosed lipid droplets or endocytosed lipoproteins, which is mediated by lysosomal acid lipase (LAL)[52]. The endocytosis of lipoproteins is facilitated by members of the LDL receptor family (mainly LDLR, vLDLR and LRP1[33,34],) in lipid rafts, or by clathrin-dependent internalization. Continuous inhibition of this key process in the fuel supply of thermogenic adipocytes in VasTg primary adipocytes may increase the observed antilipolytic effect compared to brown adipocytes that have been treated with recombinant vaspin acutely. Indeed, in addition to the LRP1[26], our screening of LDL receptor family members revealed low affinities, in part sub-nanomolar, for key receptors LDLR and vLDLR, as well as LRP2 and LRP4. Previous studies have indicated that LDLR and LRP1 do not significantly contribute to the uptake of triglyceride-rich lipoproteins in BAT[53]. However, of all the potential endocytosis receptors, vaspin bound most strongly to the vLDLR. VLDLR is significantly higher expressed in thermogenic (brown and beige) adipocytes than in white adipocytes[54], and vLDLR-mediated vLDL uptake is a crucial pathway fueling thermogenesis in BAT[55]. Therefore, binding to lipoprotein endocytosis receptors of the LDL receptor family, alongside adaptive reductions in thermogenic gene expression, likely enhances the antilipolytic effect in VasTg adipocytes compared to wildtype adipocytes acutely treated with vaspin. As this is the first report of vaspin binding to different members of the LDL receptor family, there are many avenues of research to be pursued to investigate the cellular consequences of individual receptor binding events. These will likely affect not only adipocytes, but also liver and vascular lipid uptake.

In seemingly contrasting observations, others and we found that vaspin transgenic mice were less responsive to high fat diets (HFD) and had significantly better metabolic health[25]. In our VasTg mice, this was linked to increased energy expenditure, independent of locomotor activity and food intake[28]. However, in line with the acute inhibitory effects of vaspin on BAT function in vitro and in vivo, energy expenditure is lower in VasTg mice fed a standard diet and housed at 23 °C (Supplementary Fig. 1C-D and ref. 28). The contradictory effects of vaspin on energy balance can be reconciled by considering its distinct direct/acute and indirect/long-term actions. In the present study, we demonstrate that vaspin directly inhibits adrenergic activation of lipolysis, thereby restricting the supply of fatty acid for thermogenesis in BAT under acute metabolic demands such as cold exposure or fasting. However, in the context of chronic HFD feeding, the predominant effect of vaspin is its anti-inflammatory action, which is linked to its serpin function and the inhibition of the inflammatory protease KLK7[20–22,25]. Improving AT inflammation may preserve the function of thermogenic AT, and this protective effect likely outweighs the acute inhibitory effect on lipolysis. This results in improved whole-body energy expenditure and metabolic health[56]. Furthermore,

adaptive and compensatory mechanisms may contribute to maintaining energy homeostasis in vaspin-overexpressing models. For instance, enhanced insulin sensitivity and glucose utilization in BAT and skeletal muscle could provide alternative substrates for thermogenesis and energy expenditure under HFD conditions. Further research is needed to delineate these compensatory pathways and their relative contributions to energy metabolism under various physiological and dietary conditions. Experiments studying the effects of acute cold-exposure in HFD-fed obese VasTg mice may enable the acute anti-thermogenic/anti-lipolytic effects of vaspin to be separated from the long-term anti-inflammatory effects in brown and white adipocytes.

Notably, microarray analysis of our VasTg mice revealed *Rcan2* to be the most differentially expressed gene in BAT, which was confirmed to be specific to all AT in VasTg mice and independent of the thermogenic stimulus. Importantly, our in vitro and in vivo experiments using recombinant vaspin in WT adipocytes or mice show that the acute vaspin effect and of impaired thermoregulation phenotype in VasTg mice are independent of and not mediated by alterations in *Rcan2* expression. Nevertheless, the high AT-specific expression of *Rcan2* in VasTg mice is interesting with respect to their metabolic phenotype. Rcan2 is one of three members of the regulator of calcineurin (*Rcan*) genes[57], and localized in the mitochondria (www.proteinatlas.org[58]). Its target calcineurin has many substrates including key thermogenic transcription factor NFAT, thereby controlling expression of numerous metabolic genes[59]. Global knockout of *Rcan1* and *Rcan2* both prevent age- and/or HFD-induced weight gain, though by affecting different aspects of energy homeostasis. Loss of *Rcan1*, in addition to various other organ-specific effects, has been linked to enhanced white adipocyte beiging and energy expenditure[60]. *Rcan2* KO mice in contrast are protected due to a leptin-independent reduction in food intake, with lack of brain *Rcan2* expression precluding its orexigenic effect[60]. Therefore, the function of RCAN2 in adipocytes warrants further investigation, given that VasTg mice were resistant to diet-induced obesity under HFD conditions despite high level expression of the putative orexigenic factor in AT[28].

We and others have previously demonstrated the expression and functional activity of vaspin in the brain[61,62]. Acute intracerebroventricular injection of vaspin reduces blood glucose levels via vagus-nerve-mediated suppression of gluconeogenesis in the liver[62,63]. Furthermore, central vaspin exhibits acute anorexigenic effects that may be mediated by altered expression of hunger-regulating neuropeptides or on by stabilizing anorexigenic insulin action[61,62]. Therefore, it is possible that central vaspin effects are involved in controlling BAT activity in VasTg mice. However, the inhibition of BAT thermogenesis following acute i.p. injection of vaspin in WT mice did not suggest a significant central signaling component, and our in vitro studies have demonstrated cell-autonomous effects on thermogenic adipocytes.

Vaspin therefore adds to the BATokine repertoire of molecular thermogenic brakes, which also includes the kininogen 2 - bradykinin B1/2 receptor system[64], endothelin - endothelin receptor A/B system[65], and sLRP11[66]. Interestingly, the cold-induced low- and high-molecular-weight kininogen 2, which is secreted from activated brown adipocytes, is processed by the tissue kallikrein (KLK1) to yield the active bradykinin peptide. This peptide suppresses BAT thermogenesis by binding to B1/B2 receptors[64]. Vaspin is a specific inhibitor of two kallikrein family members in KLK7 and KLK14[19,67], and although vaspin does not inhibit KLK1[68], KLK7 exhibits efficient kininogenase activity (releasing des-Arg9-bradykinin[69],), suggesting that vaspin may interact with the local kallikrein-kinin pathway to regulate the activity of thermogenic adipocytes (Fig. 5).

Finally, impaired thermoregulation in VasTg during fasting suggests that vaspin inhibition of adrenergic lipolysis may not only be specific to brown adipocytes but may also (co)regulate white adipocyte lipolysis. During fasting and cold exposure, brown adipocyte lipolysis is dispensable for maintaining physiological adaptive thermogenesis. However, adrenergic activation of WAT lipolysis is essential to provide fatty acid substrates (especially acylcarnitines subsequently generated in the liver) to fuel BAT thermogenesis[35], as well as skeletal muscle shivering thermogenesis under conditions of impaired BAT thermogenic activity. During fasting, WAT lipolysis is further accelerated by the increased presence of fasting hormones, such as glucocorticoids and glucagon, in the bloodstream, as well as by lower levels of insulin-mediated inhibition. After cold exposure, however, a WAT-lipolysis-dependent increase of systemic insulin is essential for proper BAT thermogenesis[70]. Although we did not measure insulin levels after the fasting period, but VasTg mice exhibit significantly lower insulin levels than littermate controls[28], which further supports the inhibition of WAT lipolysis by vaspin. Finally, our in vitro data using mouse and human primary white adipocytes confirmed the inhibitory effect of endogenous or recombinant vaspin on the adrenergic activation of white adipocyte lipolysis. Furthermore, the negative correlations in human SAT between the vaspin and genes related to the TCA cycle and oxidative phosphorylation support the idea that vaspin's association with reduced mitochondrial activity can also be applied to human WAT.

Together, in this work we have identified vaspin as part of the auto- and paracrine regulation of BAT thermogenesis, controlling energy expenditure in response to adrenergic stimuli. These thermogenic brakes fine-tune the activation of BAT thermogenesis, preventing excessive energy expenditure and maintaining energy homeostasis while protecting lipid and glucose stores. This discovery could lead to the development of drugs for treating obesity by releasing the brakes on BAT activity to increase energy expenditure and eliminate excess blood metabolites.

## Methods

### Study approval
**Animal studies.** All animal experiments were approved by the local authorities of the Free State of Saxony, Germany (Landesdirektion Leipzig: TVV39/14; TVV26/16, T09/21), as recommended by the responsible local animal ethics review board.

### Leipzig Obesity BioBank (LOBB)
Written informed consent was obtained from all patients. All studies were approved by the Ethics Committee of the University of Leipzig (approval numbers: 159-12-21052012 and 017-12ek) and performed in accordance with the Declaration of Helsinki, the Bioethics Convention (Oviedo), and EU Directive on Clinical Trials (Directive 2001/20/EC). All AT donors have been informed of the purpose, risks and benefits of the biobank. Ethical guidelines and EU legislation for privacy and confidentiality in personal data collection and processing is being followed, in particular directive 95/46/EC.

### Materials
Expression, purification and confirmation of biological activity of recombinant vaspin expressed in *E.coli* were carried out as previously described[71]. Recombinant glycosylated human vaspin expressed in HEK cells was from Biolegend. Chlorpromazine (CPZ), Ro 20-1724 (Ro), Brefeldin A (BFA), CL316,243 (CL), isoproterenol (ISO), forskolin (FSK), 3-isobutyl-1-methylxanthine (IBMX), norepinephrine (NE), and pertussis toxin (PTX) were from Sigma. Cilostamide (Cilo) was from Santa Cruz.

ELISA-based binding assay: Clear bottom 384-well plates (high-binding #781061, Greiner, Kremsmünster, Austria), nonfat dry milk (#T145.3, CarlRoth, Karlsruhe, Germany), anti-tetramethylrhodamine (TRITC) antibody (#A6397, Invitrogen, Waltham, MA, USA), HRP-conjugated rabbit secondary antibody (#7074S/#7076S, Cell Signaling, Danvers, MA, USA), 3,3′,5,5′-tetramethylbenzidine substrate (#ES022, Merck, Darmstadt, Germany). Recombinant human LDLR,

vLDLR, LRP2 cluster 3, LRP4, LRP6, LRP10 (#2148-HP, #8444-VL, #9578-LR, #9579-LR, #5948-LR, #1505-LR, #10647-LR, R&D, Minneapolis, MN, USA) and LRP5 (#17048, SinoBiological, Beijing, China). All recombinant receptor proteins were suitable for ligand binding studies according to the manufacturer. Recombinant human vaspin was labeled as previously described with labeling efficiency ranging from 0.8-1.0[26].

## Animal studies

**Breeding and Housing.** Female C57BL/6 NTac mice (Taconic Bioscience, Lille Skensved, Denmark) and adipose-tissue specific human vaspin-transgenic mice on the C57BL/6N background (VasTg, described in ref. [28]) were bred at the Sächsische Inkubator für Klinische Translation (SIKT), Leipzig. For the VasTg line, non-transgenic littermates served as controls (WT). All mice were housed in pathogen-free facilities at 23 °C, 55% humidity, on a 12 h light/dark cycle, were fed a standard chow diet (V1534, 9 kJ% from fat, Ssniff®, Soest, Germany) with ad libitum access to food and water, except when indicated. At 11 weeks of age, mice were adapted to single housing in rodent climate chambers (HPP750 life, Memmert, Germany or MKKL1200, Flohr Instruments, Netherlands) for at least 3 days.

## Acute cold exposure

For cold exposure experiments, single-housed (23 °C), ad libitum fed mice were transferred into a climate chamber set to 8 °C and remained there for 6 h, with or without access to food. Thermal imaging and measurements of blood glucose were done at the beginning and at the end of the experiment. Rectal body temperature was measured hourly. C57BL6/N mice received an intraperitoneal (i.p.) application of vaspin (1 mg/kg in PBS) or PBS 2 h before cold exposure.

## Fasting-feeding transition

Single-housed (23 °C), ad libitum fed mice were fasted starting at 6:00 pm (dark phase) and were refed 16 h after food withdrawal. Thermal imaging and rectal body temperature measurements were done before and after fasting, as well as 8 h after refeeding.

## Body temperature measurements and thermal imaging

Rectal body temperature was measured using a special probe (TH-5 Thermalert Clinical Monitoring Thermometer, Physitemp Instruments). BAT and tail surface temperatures were measured by infrared thermal imaging (VarioCAM® hr, Infratec, Dresden, Germany) as previously described[72].

## Organ and serum collection

After each experiment, animals were sacrificed, serum and organs collected and processed for histological and biochemical analyses or snap frozen in liquid nitrogen. Serum free fatty acid levels were measured using NEFA-HR(2) Assay (Fujifilm Wako Chemicals Europe) according to the manufacturers' protocol.

## Histology

AT histology was performed as previously described[22].

## Cell culture

**Adipocyte culture and differentiation.** Primary brown adipocytes from male and female C57BL/6 N mice, VasTg mice or WT littermates, as well as immortalized brown adipocytes (imBA[73],) were cultured and differentiated as previously described[22]. Reverse transfection of imBA cells for siRNA-mediated knockdown (siLrp1 #s69313 and control #4390843, Ambion, Kassel Germany) was performed as previously described[26].

## Proteome and phosphoproteome analysis

For phosphoproteome analysis, imBA cells were treated with 500 ng/ml vaspin for 30 min. For proteomics, imBA cells were treated with 500 ng/ml vaspin for 6 h. Sample preparation, Tandem mass tag (TMT) labeling or phospho-peptide enrichment, UPLC-MS/MS analysis, data acquisition, processing and statistical analysis are described in detail in the **Supplementary Methods**.

## Analyses of signal transduction

Differentiated adipocytes were serum starved for 3 h and treated with or without adrenergic agonists norepinephrine, isoproterenol or CL316,243 (NE: 1 μM / 15 min; ISO: 100 nM / 15 min; CL: 1 μM / 15 min), adenylyl cyclase agonist forskolin (FSK: 1 μM / 15 min), pertussis toxin (PTX: 100 ng/mL / 18 h prior to further treatment), 3-Isobutyl-1-methylxanthin (IBMX: 25 μM / 5 min prior to further treatment), Cilostamide (Cilo: 10 μM / 5 min prior to further treatment), Ro 20-1724 (Ro: 10 μM / 5 min prior to further treatment), or RAP (1 μg/ml / 20 min prior to further treatment), respectively, prior to Western blot analysis. To study vaspin effects on cellular signaling and metabolism, 500 ng/ml of recombinant human vaspin was added to cells for 5-20 min prior stimulation, except when using primary brown adipocytes from VasTg mice. To assess the necessity of vaspin secretion, VasTg primary brown adipocytes were treated with BFA (20 μM) during the 3 h starvation and culture medium was exchanged before stimulation of lipolysis to further remove residual vaspin in the supernatant. Following stimulation, cells were washed twice with PBS and stored at -80 °C until further analyses.

## Oxygen consumption assays

Primary brown adipocytes were seeded into 24-well V28 plates (22,000 cells per well), differentiated as described above and Cell Mito Stress Tests were performed in a Seahorse XFe24 as previously described[74].

## In vitro lipolysis and lactate assays

Lipolysis in differentiated adipocytes was assessed after overnight serum starvation, with or without addition of 500 ng/ml vaspin for 30 min prior measurements under basal conditions and after addition of CL, FSK or ISO (final concentrations 1 μM or 100 nM) for 1–2 h using standard assays according to the manufacturers' protocol (assay medium 2% free fatty acid free BSA in DMEM; NEFA-HR(2) Assay, Fujifilm Wako Chemicals Europe). The L-Lactate Assay Kit (#ab65331) was used to assess lactate release in supernatants from lipolysis assays.

## SDS-PAGE, Western blot and ELISA

Preparation of cell and tissue lysates, SDS-PAGE, Western blot and antibody incubations were performed as previously described[14]. Chemiluminescence was detected in the G:BOX documentation system (Syngene, Cambridge, UK) followed by densitometric quantification using the GeneTools software (Syngene). The following primary antibodies were used: from Cell Signaling Technologies: phospho-PKA substrates (RRXS*/T*, #9624), HSL (#4107), phospho-HSL (Ser660, #4126), p38 MAPK (#9212), phospho-p38 MAPK (Thr180/Tyr182, #4511), anti-rabbit-HRP (#7074), anti-mouse-HRP (#7076); from Abcam: UCP1 (ab10983); from Sigma-Aldrich, St. Louis, MO, USA: ACTB (#A1978); from Thermo Fisher Scientific: OXPHOS (#45-8099). Human vaspin in serum or cell supernatants were measured by ELISA (human vaspin, Adipogen).

## Quantitative real-time-PCR (qPCR)

RNA from BAT was isolated using RNeasy Lipid Tissue Mini kit (Qiagen, Hilden, Germany) as specified by the manufacturer. qPCR was performed using the LightCycler System LC480 and LightCycler-DNA Master SYBR Green I Kit (Roche, Mannheim, Germany). Adipocyte gene expression was calculated by ΔΔCT method and normalized to *Nono, Hprt* or *36b4* levels[75], as indicated. Primer sequences are listed in (Supplementary Table 2).

### Micro array analysis of BAT gene expression

Microarray analysis (Clariom S Assay, mouse, Thermo Fisher) of BAT gene expression of acutely cold-exposed female WT and VasTg mice (6 h at 8 °C, $n = 3/4$ per group) was conducted at the Core Unit DNA Technologies (Core Facilities of the Faculty of Medicine; University of Leipzig) and data processing and analysis was done as previously described[74]. Only differentially expressed genes with $p$ value < 0.05 were considered.

### ELISA-based binding assay

Binding of vaspin to LDL-superfamily receptors LDLR, vLDLR, LRP2-C3, LRP4, LRP5, LRP6 and LRP10 were assessed using an ELISA-based approach, as previously described[26,76]. Briefly, 384-well clear-bottom plates were coated with recombinant receptor proteins at a concentration of 500 ng/mL in Tris-buffered saline (TBS) containing 5 mM $CaCl_2$ (pH 8.0). Following coating, wells were blocked with 5% nonfat dry milk in TBS for 2 h at room temperature. Increasing concentrations of vaspin was then added to individual wells and incubated for 2 h at room temperature. Binding detection was performed using primary antibodies specific to TAMRA for 1.5 h, followed by a 1-h incubation with HRP-conjugated rabbit secondary antibodies. The reaction was developed using the HRP substrate 3,3',5,5'-tetramethylbenzidine, then stopped with 0.16 M $H_2SO_4$. Absorbance was measured at 450 nm using a FlexStation 3 multimode microplate reader (Molecular Devices, San Jose, CA, USA). Each interaction was analyzed in at least four technical replicates. Absorbance values were plotted against log(M), and $EC_{50}$ values were determined by non-linear regression analysis (log[agonist] vs. response [three parameters]) using Prism 10 (Graph-Pad Software, Boston, MA, USA).

### Human studies

**The Leipzig Obesity BioBank Cohort (LOBB).** The human cross-sectional cohort of the Leipzig Obesity BioBank (LOBB; https://www.helmholtz-munich.de/en/hi-mag/cohort/leipzig-obesity-bio-bank-lobb) comprises 1143 patients with obesity (women: $n = 813$; men: $n = 330$; mean ± standard deviation: age = 46.8 ± 11.8 years, BMI = 49.1 ± 8.4 kg/m$^2$) and 26 patients without obesity (women: $n = 14$; men: $n = 12$; age = 58 ± 13.3 years, BMI = 25.8 ± 2.5 kg/m$^2$). Samples were collected during elective laparoscopic abdominal surgery between 2008 and 2018, as previously described[77,78]. Exclusion criteria encompassed participants under 18 years of age, chronic substance, or alcohol misuse, smoking within the 12 months prior to surgery, acute inflammatory diseases, use of glitazones as concomitant medication, end-stage malignant diseases, weight loss exceeding 3% in the three months preceding surgery, uncontrolled thyroid disorder, and Cushing's disease.

### Bulk RNA sequencing, analysis and Gene Set Enrichment Analyses (GSEA)

Bulk RNA sequencing and data processing was done as previously described[78]. To predict the gene function of *SERPINA12* and identify potential associations with Kyoto Encyclopedia of Genes and Genomes (KEGG) pathways[79], we conducted Gene Set Enrichment Analysis (GSEA[80],) of the RNA sequencing data from the LOBB cross-sectional cohort using CorrelationAnalyzeR (v1.0.0[81];). The single gene mode R package was employed, considering the co-expression correlations of SERPINA12 with other genes in the dataset. Genome-wide Pearson correlations were employed as a pre-ranking metric for the GSEA algorithm. The correction for multiple testing was performed using the false discovery rate (FDR) method.

### Human subcutaneous adipose tissue collection, isolation of the stroma vascular fraction and differentiation of adipocytes

SAT samples were collected during elective esthetic and post-bariatric surgery at the Division of Plastic, Esthetic and Special Hand Surgery of University Hospital Leipzig. Collection sites were predominantly the lower abdomen during abdominoplasty as well as medial thigh during thighplasty and lower back during dorsal body lift. All operations were performed under general anesthesia, use of local anesthetics excluded patients from enrollment in the study. Patients with prior liposuction or cryolipolysis to the respective area were excluded from sample collection. Electrocautery was used to prepare the subcutaneous tissue for resection. Thermally damaged tissue and skin were removed using scissors or scalpel, and fat samples were placed into sterile sample containers for immediate processing.

AT samples were washed, connective tissue was removed and the fat was processed with scissors to obtain a homogenous mixture, that was digested with 500 units collagenase (Type II, Gibco) per gram of tissue in adipocyte isolation buffer (100 mM HEPES, 123 mM NaCl, 5 mM KCl, 1.3 mM $CaCl_2$, 5 mM Glucose, 1% ZellShield®, 4% BSA) for 45 min at 37 °C. Digested fat was passed through a 300 μm syringe strainer (pluriSelect®).

Following centrifugation, the SVF containing infranatant was collected and processed separately from the supernatant which contained the mature adipocytes. The mature adipocytes were extensively washed with buffer and free fat was carefully removed. The adipocytes were seeded on transwells (Corning, New York, USA) and cultivated as membrane mature adipocyte aggregate culture as described previously[82].

The SVF was washed twice with buffer and the pellet dissolved in red blood cell lysis buffer and incubated for 7 min at RT, before adding 10 mL of DMEM/F12 followed by further 5 min of incubation. The suspension was passed through a 30 μm MACS® SmartStrainer (Miltenyi Biotec) and preadipocytes were pelleted by centrifugation at 500 g for 10 min followed by resuspension in growth medium (DMEM/F12 + 10%FCS + 1% Zellshield®). Cells were seeded in 12- or 96-well plates at densities of 300.000 or 30.000 cells per well, respectively. Preadipocytes were cultivated in growth medium until confluency (day 0, D0) and subsequently differentiation was induced by changing to adipogenic medium (with 10% fetal calf serum) until D16, with medium changes every 48 h, as previously published[27].

### Immunofluorescence of human mature SAT adipocytes

Adipocytes were washed with PBS and a 1:8 adipocyte suspension in growth medium was prepared. TAMRA-labeled vaspin was added to the cell suspension to a final concentration of 100 nM and incubated for 30 min at 37 °C and 5 % $CO_2$ with continuous rotation. Subsequently, cells were centrifuged and washed twice with 1 M NaCl and fixed with 10% formaldehyde and permeabilized with 0.1% TRITON X-100. Unspecific binding sites were blocked with 10% normal goat serum and cells were incubated overnight at room temperature with anti-LRP1 antibody (ab92544, abcam, Cambridge, UK). The next day, cells were washed and incubated with goat anti-rabbit DyLight(TM) 488 antibody (VEC-DI-1488, Biozol, Hamburg, Germany). Cells were analyzed using a BZ-X800 Microscope (Keyence, Osaka, Japan).

### Statistical analyses

Data are presented as means ± SEM. Statistical analyses were performed using GraphPad Prism 10 (GraphPad, San Diego, CA, USA). Methods of statistical analyses were chosen based on the design of each experiment and are stated in the figure legends. If not stated otherwise, adj. $p < 0.05$ were considered statistically significant.

### Reporting summary

Further information on research design is available in the Nature Portfolio Reporting Summary linked to this article.

## Data availability

All the necessary data to evaluate the conclusions in the paper are provided in the paper itself and/or the Supplementary Materials and

source data are provided with this paper. Microarray data have been deposited in the ArrayExpress database at EMBL-EBI[83] under accession number E-MTAB-14068. The mass spectrometry proteomics data have been deposited to the ProteomeXchange Consortium via the PRIDE[84] partner repository with the dataset identifiers PXD068908, PXD069112 and 10.6019/PXD069112. Access to the human RNA-seq data from the LOBB study is regulated by the LOBB steering committee and requires an approved Data Use Agreement (DUA) outlining the permitted research purposes and protections for participant privacy. Requests will be acknowledged within 5–10 business days and, following review, granted or denied typically within 2–4 weeks, depending on the completeness of the application and the scope of the requested data. Access is restricted to non-commercial, non-identifiable research use. Data access may be granted to named researchers or institutions after execution of a DUA and approval by the steering committee. For access, please contact . Source data are provided with this paper.

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

## Acknowledgements

This work was funded or supported by grants of the Deutsche Forschungsgemeinschaft, project number 209933838 (SFB1052 "Obesity Mechanisms": A1 to MS, B1 to MB, B3 to PK, B4 to NK, B12 to KK, C7 to JTH, Z3 to MvB) and project number 652 530364326 (to KS). IFB AdiposityDiseases was supported by the Federal Ministry of Education and Research (BMBF), Germany, FKZ: 01EO1501. HB is supported by a doctoral scholarship of the Studienstiftung des Deutschen Volkes. We acknowledge technical support of the Core Unit DNA Technologies of the Faculty of Medicine, University of Leipzig, and thank Dr. Knut Krohn and Petra Süptitz. We thank Wenfei Sun and Hua Dong for supporting human adipose tissue RNA-seq, and Eva Böge, Jenny Schuster and Lisa Gärtner from the animal facility, and Claudia Gebhardt, Olivia Paetow, Maren Wiermann, and Nele Ferekidis for excellent technical assistance. We are grateful to all donors of adipose tissue samples.

## Author contributions

I.R., J.W. and J.T.H. conceptualized, initiated and developed the project and wrote the manuscript. I.R., J.W., K.M., H.B., A.M., K.K. and N.K. performed or assisted with experiments and analyzed data. A.H. performed bioinformatic analyses. I.Kac and D.T. contributed data and provided expertise. H.G., L.K., I.Kar, K.S. and Mv.B. performed proteomic analyses. J.B., P.K. generated the VasTg mouse line. R.N., S.L. performed esthetic and post-bariatric surgery and collected human adipose tissue samples. A.G. performed RNA-seq and pre-processing of samples from the Leipzig Obesity Biobank. C.W., M.S. and M.B. contributed human data. J.T.H. coordinated and supervised the project. All authors read, edited and approved the manuscript.

## Funding

## Competing interests

M.B. received honoraria as a consultant and speaker from Amgen, AstraZeneca, Bayer, Boehringer-Ingelheim, Lilly, Novo Nordisk, Novartis, and Sanofi. All other authors declare no conflicts of interest. The funders had no role in the design of the study, the collection, analyses, or interpretation of data, in the decision to publish the results and the writing of the manuscript.
