## [Transparent Peer Review file · Nature Communications]

Inhibition of adipocyte lipolysis by vaspin impairs thermoregulation in vivo.

Corresponding Author: Dr John Heiker

Version 0:

Reviewer comments:

Reviewer #1

(Remarks to the Author)

In this manuscript, the authors describe a role for the secreted protein vaspin in brown adipose tissue function. Through studies in transgenic mice overexpressing vaspin in adipose tissue and studies in cells and human tissue, they present data supporting a model in which vaspin acts as a thermogenic brake. It does so by inhibiting adrenergic signaling and the activation of lipolysis, and these effects appear to be downstream of binding to the LRP1 receptor. Finally, the authors present data from a large human cohort consistent with an anti-lipolytic role for vaspin in human subcutaneous adipose tissue. This work highlights a new role for vaspin as a BATokine and is likely to be of broad interest to researchers in the field. However, a number of key concerns need to be addressed in order to fully support the conclusions made.

Major Points:

1. The methods section of this paper does not provide any details on how the vaspin transgenic mice were generated. Based on reference 24, it sounds like the authors used aP2 as a driver. Is this correct? If so, the authors should acknowledge that aP2 is not only expressed in adipocytes, which could confound some of the conclusions of this study. One way to further clarify the site of expression would be to fractionate adipose tissue from transgenic mice into mature adipocytes and stromal vascular fraction and to then measure vaspin RNA levels in each fraction.
2. The mechanistic studies are incomplete. The authors argue that the anti-thermogenic and anti-lipolytic effects of vaspin are downstream of binding to LRP1, which they argue is then internalized. While this argument would be most conclusive if the authors generated mice with brown adipocyte specific deletion of LRP1, this is beyond the scope of this manuscript. However, the authors should address the following:
 - a) This point would be more compelling if the authors performed microscopy to show the subcellular localization of vaspin upon internalization.
 - b) To further probe the requirement for internalization of vaspin in its signaling actions, the authors could do experiments with or without vaspin in control and Lrp1 knockdown cells using endocytosis inhibitors.
 - c) To further support the argument that the phosphoproteome changes induced by vaspin mediate its effects, the authors should consider mutating one of the PKA phosphorylation sites and seeing if the effects of vaspin on lipolysis and/or thermogenesis are attenuated.
 - d) Can the authors speculate at least as to how vaspin binding to Lrp1 and internalization go on to modulate PDE activity?

Minor Points:

1. In Figure 1D and 1P, glucose levels increase after cold exposure in the fasted state. This is somewhat surprising as prior studies have shown that cold exposure in fasted mice leads to an acute decrease in glucose. Can the authors clarify?
2. In figure 2, the phosphoproteomics suggests showed increased phosphorylation of PKA substrates (lines 144-147), yet the Western blot in Figure 2 showed decreased phosphorylated PKA substrates. Can the authors explain?

Reviewer #2

(Remarks to the Author)

This paper reports that the adipokine Vaspin can reduce catecholamine-stimulated adipocyte lipolysis, which is associated with reduced brown fat activity. Vaspin-overexpressing mice exhibit reduced brown fat activity and cold-intolerance associated with diminished activation of PKA-signaling. These effects are recapitulated by acute treatment of mice with recombinant Vaspin. Studies in isolated cells show that Vaspin acts directly on adipocytes via the LRP1 receptor to repress β -adrenergic-induced activation of PKA signaling and lipolysis. It is clear that high levels of Vaspin can repress lipolysis. However, there are some important issues and questions, especially related to the physiological role and significance of this pathway that should be addressed for publication in Nat Commun.

Major concerns:

1. The studies rely exclusively on overexpression/gain-of-function methods with super-physiological levels of Vaspin. The Vaspin injections produced "four order of magnitude higher than physiological levels". Cell culture studies also utilize a high dose of Vaspin, and produced modest effects. Further studies utilizing lower and more physiological doses of Vaspin should be presented. Additionally, the central conclusions could be strengthened by loss-of-function models, ie. inhibiting receptor in BAT, reducing Vaspin levels/activity?
2. Related to #1, it would be important to evaluate the levels of Vaspin (circulating and adipose tissue) in response to cold exposure, fasting, etc. It is unclear whether this pathway is regulated by physiological (or pathological) stimuli.
3. The effect of Vaspin on mitochondrial function/cell respiration is not sufficiently explained. Fig. 2K, L show that basal respiration is suppressed by Vaspin, prior to FSK treatment. Indeed the magnitude of this reduction (ie. fold reduction) appears similar to that observed after FSK treatment. This is presumably independent of lipolysis and fatty acid oxidation? This effect may contribute greatly to the in vivo phenotypes and should be assessed further.

Minor

4. Fig 4J. This analysis is not clearly described. Are these genes/pathways that correlate with higher vs. lower vaspin expression in human fat tissue? How was this done?
5. It should be clearly stated that the transgene is driven from the aP2/Fabp4-promoter, which is known to have expression in many tissues, including in the brain from early development.

Reviewer #3

(Remarks to the Author)

The present manuscript by Rapöhn et al. reported that vaspin or SEPINA12, an adipocyte-derived serine protease inhibitor, represses BAT thermogenesis by attenuating adrenergic signaling and lipolysis in brown adipocytes. The authors proposed a model in which vaspin binds to the LRP1 receptor on the cell surface and stimulates PDEs, resulting in reduced cAMP-PKA signaling and impaired brown fat thermogenesis. However, several major issues are identified.

1. A previous study by the authors' group reported that transgenic mice expressing vaspin in adipose tissues, driven by the aP2 promoter, limited high-fat diet-induced obesity, improved glycemic control, and elevated whole-body energy expenditure (PMID: 37152954). In contrast, the present work showed that vaspin overexpression attenuates brown fat thermogenesis. It is unclear how these apparently contradictory results can be reconciled because impaired lipolysis would lead to reduced fatty acid oxidation in brown fat, skeletal muscle, and other organs. If a compensatory mechanism exists, the authors need to address such mechanisms.
2. The authors used the aP2 promoter to drive vaspin in their transgenic mice. It is now well recognized that the aP2 promoter is expressed in many tissues other than adipocytes, and as such, it is unclear how vaspin acts on adipocyte lipolysis as a secretory molecule or acts intracellularly. Even though the authors performed cultured cell experiments, the dose used in the study (0.5 μ g/ml) was far higher than a physiological dose.
3. Related to the above point, the efficacy of PKA blots appeared potent when transgenic mice-derived adipocytes were used compared to ones with purified vaspin (e.g., Fig2D, E). It is conceivable that vaspin acts intercellularly rather than the LRP1-mediated signaling pathway.
4. For the PKA data in Fig.3D, G, J, Fig.4F, and 4M, the effect sizes were low, raising a concern about the robustness of the experiments.
5. The authors proposed a model in which vaspin acts on PDEs by using IBMX, but the underlying mechanism was not explored. It is possible that vaspin and PDEs are completely parallel pathways.

Version 1:

Reviewer comments:

Reviewer #1

(Remarks to the Author)

The revised manuscript has been substantially improved with the addition of new experimental data. The combination of these data and those shown in the rebuttal have addressed all of the concerns raised in my initial review. This is a nicely done study that will be of interest to people in the field.

Reviewer #2

(Remarks to the Author)

The authors have addressed many of the reviewers' previous critiques. The paper is well presented and the conclusions are generally well supported. There are a couple of remaining points that should be addressed before publication.

1. I am still unclear about the seemingly striking suppressive effect of Vaspin OE on the basal respiration of brown adipocytes isolated from mice (Figure 2K). The proteomic analyses in the paper indicates that respiratory proteins were reduced in Vaspin-treated brown adipocytes. From Results: "Among the proteins with lower abundance, pathway analysis revealed significant enrichment of proteins related to oxidative phosphorylation, such as core subunits of respiratory chain complex I (NDUFS6, NDUF3, NDUF12 and NDUF8), complex III (UQCRF1), and COX2A2, the terminal oxidase in mitochondrial electron transport (KEGG mouse 2019, Figure 2C and Supplementary Table 7)." However, there appears to be no effect on these components in vivo (Fig. S1), and as alluded to in the reviewer response - can the authors add to the discussion of this point. Is this also LRP dependent?

2. Fig 3B-F. This experiment is not very informative without having the +CL and no BFA condition. The reader can't appreciate the effect of secretion without seeing the untreated (no BFA) condition in the same experiment.

3. Determining the effect of Vaspin on FFA levels in vivo would strengthen the conclusions. For example, is there a decrease in FFA levels in vivo in response to CL administration? Or during cold exposure/fasting?

Reviewer #3

(Remarks to the Author)

The authors provided new data that addresses the reviewer's comments satisfactorily. Congratulations!

Version 2:

Reviewer comments:

Reviewer #2

(Remarks to the Author)

The authors have addressed the remaining questions. This is a well executed study that will be of substantial interest to the field.

Rebuttal letter to the manuscript: *"Inhibition of adipocyte lipolysis by vaspin impairs thermoregulation in vivo."*

We thank the editor and all the referees for their careful assessment of our work and their constructive comments, questions, and suggested experiments/analyses.

We were very happy to read that the reviewers generally acknowledge the novelty and significance of our study, particularly in identifying vaspin as a BATokine that inhibits adrenergic lipolysis via LRP1-mediated signaling. The reviewers also appreciated the integration of mouse models, cellular studies, and human cohort data, emphasizing the broad relevance of the findings.

However, they also voiced so concerns particularly with regard to the transgenic mouse line, the more pronounced effects in transgene expressing adipocytes compared to wildtype cells treated with recombinant vaspin, and related to the underlying molecular mechanism and involvement of the LRP1.

Here, we have now addressed all the referees' comments and suggestions in full and believe that the manuscript has significantly improved.

In summary, we made these major changes and additions:

- We addressed the adipocyte specific expression of the transgene and demonstrate lack of leaking expression in other tissues.
- We provide for reviewers only pilot data from BAT-specific LRP1-KO mice (using Ucp1-cre). We show that in contrast to previously published Fabp4-Cre mediated LRP1-KO in WAT, these mice so not show dysfunctional AT development, and present Seahorse data showing increased adrenergically stimulated mitochondrial respiration in LRP1-KO primary brown adipocyte and indirect calorimetry showing increased whole body energy expenditure in LRP1-KO mice.
- We have performed additional experiments to establish involvement of the LRP1. We first show that vaspin needs to be secreted (using BFA) and use siRNA for knock down, endogenous ligand RAP to reduce cell surface LRP1 as well as chlorpromazine to prevent internalization after vaspin binding. All conditions prevented vaspin's inhibitor effect on lipolysis.
- We investigated binding to numerous LDL receptors and class B scavenger receptors, finding low/sub nanomolar affinity for LDLR, VLDLR as well as LRP2 and 4, in addition to previously shown LRP1. The interaction of vaspin with these receptors would block lipoprotein endocytosis to fuel thermogenesis (as shown for the VLDLR) and may represent an additional mechanism of controlling lipid turn over.
- We clarified and updated the Methods, and we provide a new table with the receptor binding data.

We have prepared a revised version of the manuscript by carefully considering all the reviewer's comments and suggestions. With these changes, we hope the reviewers share our

enthusiasm for this work and we resubmit our manuscript for a secondary evaluation and hope that it is now matching the high standards of Nature Communications.

In detail, we would like to respond to the reviewer's comments as follows.

COMMENT: Several Figures in the rebuttal letter are replicated from the main manuscript and placed directly with the answers to improve readability. We have clearly indicated where the same data is presented in the revised manuscript and sometimes even kept the labels of the panels from the manuscript to make the identification very clear.

We have also updated the numbering of Figures and Tables in the Manuscript and refer in the rebuttal letter always to the new revised numbering scheme.

Reviewers' comments:

Reviewer #1 (Remarks to the Author):

In this manuscript, the authors describe a role for the secreted protein vaspin in brown adipose tissue function. Through studies in transgenic mice overexpressing vaspin in adipose tissue and studies in cells and human tissue, they present data supporting a model in which vaspin acts as a thermogenic brake. It does so by inhibiting adrenergic signaling and the activation of lipolysis, and these effects appear to be downstream of binding to the LRP1 receptor. Finally, the authors present data from a large human cohort consistent with an anti-lipolytic role for vaspin in human subcutaneous adipose tissue. This work highlights a new role for vaspin as a BATokine and is likely to be of broad interest to researchers in the field. However, a number of key concerns need to be addressed in order to fully support the conclusions made.

We thank this reviewer for his positive comments on our work and have addressed his concerns as follows.

Major Points:

1. The methods section of this paper does not provide any details on how the vaspin transgenic mice were generated. Based on reference 24, it sounds like the authors used aP2 as a driver. Is this correct? If so, the authors should acknowledge that aP2 is not only expressed in adipocytes, which could confound some of the conclusions of this study. One way to further clarify the site of expression would be to fractionate adipose tissue from transgenic mice into mature adipocytes and stromal vascular fraction and to then measure vaspin RNA levels in each fraction.

We had previously reported tissue specific expression of the human vaspin transgene in VasTG mice (Rapöhn et al. *Frontiers in Endocrinology* 2023). We have now repeated analysis of transgene expression (human vaspin / SERPINA12) in a tissue panel of VasTG mice showing exclusive expression in AT depots (Reviewer Figure 1A). As also reported for Prdm16Tg Prdm16xFabp4 mice (PMID: 29131158), we did not observe transgene expression in the brain. We furthermore fractionated adipose tissue of VasTG mice into mature adipocytes and stromal vascular fraction (SVF) and quantified human vaspin mRNA expression in each fraction. Our data confirm that vaspin is specifically expressed in adipocytes (Reviewer Figure 1B), minimizing concerns about non-adipocyte contributions. Consistently, in we observed the inhibitory vaspin effect on lipolysis in our in vitro experiments using isolated primary SVF-

derived brown and white adipocytes from VasTG mice showing inhibited lipolytic response after adrenergic stimulation.

We nevertheless acknowledge the limitations of the α 2 promoter in the Discussion.

Reviewer Figure 1: A) *SERPINA12*/ human vaspin gene expression in BAT and WAT depots, as well as brain, kidney, liver and muscle of VasTG mice demonstrates AT-specific transgene expression. B) Expression of adiponectin, *Fabp4*, mouse and human vaspin (*Serpina12* and *SERPINA12*, respectively) in the adipocyte and SVF fraction of iWAT from VasTG (red circles) and littermate controls (black circles) demonstrates adipocyte specific expression of *Fabp4* and the human vaspin transgene. Data are shown as mean \pm SEM. Statistical significance was evaluated by 2-way ANOVA and Tukey's posttest. * p -value < 0.05, ** p -value < 0.01, *** p -value < 0.001. **A and B** are shown as **Supplementary Figures 1A-B**.

2. The mechanistic studies are incomplete. The authors argue that the anti-thermogenic and anti-lipolytic effects of vaspin are downstream of binding to LRP1, which they argue is then internalized. While this argument would be most conclusive if the authors generated mice with brown adipocyte specific deletion of LRP1, this is beyond the scope of this manuscript.

We fully agree and have in parallel generated BAT-specific (*Ucp1*-Cre driven) LRP1-BAT-KO mice. We only have preliminary data, but these are consistent with a regulatory function of LRP1 in brown adipocytes (**Reviewer Figures 2-4**).

First, we find that primary brown adipocytes of LRP1-BAT-KO show full differentiation capacity and similar lipid incorporation as wild-type cells (**Reviewer Figure 2A-B**). While in vitro *Lrp1* knockdown was significant, but only roughly 25% (**Reviewer Figure 2C**), thermogenic genes such as *Ucp1*, *Adrb3*, and *Cidea* were significantly higher expressed in LRP1-BAT-KO adipocytes (**Reviewer Figure 2D**). Focusing regulation of lipolysis, we observed similar, rather lower FSK-induced PKA activation (**Reviewer Figure 2E-F**), however, Seahorse assays revealed significantly higher respiratory response to FSK and higher maximal respiration in fully differentiated primary brown adipocytes of LRP1-BAT-KO mice (**Reviewer Figure 2G-H**). We observed the same results when stimulating adipocytes with NE (not shown). These results are consistent with our observation of higher lipolysis in CL-treated imBA cells after *siLrp1* knock-down (**Figure 3J**).

Figure Redacted

Notably, previously generated adipocyte-specific LRP1-AT-KO mice (using the Fabp4/Ap2-Cre) had significantly lower body weight and fat mass due to impaired lipid incorporation and adipocyte dysfunction. Male and female LRP1-BAT-KO mice only show a trend for lower body weights and otherwise have normal body length, body temperature and in trend lower blood glucose levels (**Reviewer Figure 3A-D, F-I**) and importantly, they show no differences in AT mass (eWAT, iWAT and BAT; **Reviewer Figure 3E, J**)

Figure Redacted

Consistent with a regulatory role of LRP1 in mature brown or thermogenic adipocytes, first data on energy expenditure at room temperature in mice, selected for similar lean and fat mass (**Reviewer Figure 4A-B**), showed significantly higher energy expenditure during the active (dark) phase (**Reviewer Figure 4C-D**).

Figure Redacted

These preliminary data are in line with a regulatory and likely suppressive function of the LRP1 in mature brown or thermogenic adipocytes.

However, as these are preliminary data and ongoing studies, we would prefer to not include them in this manuscript but wanted to share them with the reviewers.

However, the authors should address the following:

a) This point would be more compelling if the authors performed microscopy to show the subcellular localization of vaspin upon internalization.

In our recent paper identifying the LRP1 as the endocytosis receptor for vaspin (Tindall et al. FEBS J 2023), we have already used fluorescence microscopy to show vaspin vesicular/endosomal uptake, subsequent lysosomal localization and degradation hours after internalization (**Reviewer Figure 5**), while the LRP1 seems to be recycled to the cell surface. However, repeating fluorescence microscopy in human mature and SVF-derived adipocytes using TAMRA-labeled vaspin and by performing cellular fractionation followed by Western blot analysis, we observed that part of internalized vaspin is translocated to the nucleus, prompting investigation of vaspin DNA binding. We found that vaspin is a high affinity ($K_d < 100\text{nM}$) non-specific DNA-binding serpin and this work is currently in revision.

Reviewer Figure 5: A) Fluorescence microscopy showing lysosomal localization of TAMRA-labeled vaspin after internalization in 3T3-L1 cells, B) which can be stabilized using chloroquine (Figure 8 A and D from Tindal et al. FEBS J 2023). C) Fluorescence Microscopy from human mature SAT adipocytes incubated with TAMRA_labeled vaspin and Western blot analysis after cellular fractionation (from Möhli et al., in revision).

b) To further probe the requirement for internalization of vaspin in its signaling actions, the authors could do experiments with or without vaspin in control and Lrp1 knockdown cells using endocytosis inhibitors.

We thank this reviewer for these suggestions and have performed additional experiments to address whether vaspin needs to be secreted and acts extracellularly (using BFA, Reviewer Figure 6A-E) and whether the inhibitory effect on lipolysis requires cell surface LRP1 (using RAP pretreatment) and is dependent on internalization (using chlorpromazine, cpz) (Reviewer Figure 6M-N).

We found that blocking of protein secretion using BFA abolished the inhibitory effect in primary brown adipocytes from VasTg mice. We then pretreated imBA with RAP, to induce ligand-bound internalization and lower LRP1 receptor density on the cell surface before analyzing the effect of vaspin on CL-induced lipolysis. As already shown, RAP treatment (and interaction with LRP1) alone did not affect PKA-activation and lipolysis but did prevent vaspin-mediated inhibition of lipolysis. This was also observed for inhibition of clathrin-mediated internalization, and cpz prevented vaspin inhibition of CL-induced lipolysis.

Reviewer Figure 6: A) Regulatory mechanisms of adrenergic signaling and levels of intervention (created with Biorender). B) Cell culture supernatant of human vaspin (SERPINA12) after 3 h starvation (control) and parallel blocking of vaspin secretion using BFA (BFA – after 3 h prior medium change, final – post stimulation with CL) before signal transduction and lipolysis assays in differentiated primary brown adipocytes from VasTg measured by ELISA (n = 3 per condition). C-E) Western blot analysis and

D) quantification of basal and CL-induced PKA-activation as well as E) free fatty acid release in differentiated primary brown adipocytes from VasTg and WT mice after blocking vaspin secretion (NEFA, $n = 3$ per condition). M) LRP1 ligand RAP: Western blot analysis of basal and CL-induced PKA-activation in RAP or vaspin treated differentiated imBA. N) Blocking of vaspin binding to LRP1 (RAP preincubation) or LRP1 internalization by clathrin-mediated endocytosis (using CPZ): Quantification of basal and CL-induced free fatty acid release with or without vaspin treatment in differentiated imBA ($n = 19-20$ per condition). Data are presented as mean \pm SEM of at least two (B-E, M) independent experiments. Statistical significance was evaluated by one-way ANOVA with Šídák's (B, D-E) post-hoc or uncorrected Fischer's LSD (N). * p -value < 0.05 , ** p -value < 0.01 , *** p -value < 0.001 .

We have included this new data in the revised manuscript in Figure 3.

c) To further support the argument that the phosphoproteome changes induced by vaspin mediate its effects, the authors should consider mutating one of the PKA phosphorylation sites and seeing if the effects of vaspin on lipolysis and/or thermogenesis are attenuated.

We agree that this is an interesting line of thought and would be an interesting experiment. The increased phosphorylation at PKA residue Ser83 identified by phosphoproteomics in vaspin-stimulated brown adipocytes has recently been linked to STK25 kinase activity and shown to induce an inactive conformation. This inhibitory effect is consistent with our finding of an inhibitory effect of vaspin on lipolysis. However, since it represents only one of nearly 100 differentially phosphorylated sites found in kinases/proteins with associated kinase activity, and since our in vitro results indicate that modulation of PDE activity by vaspin underlies or at least contributes substantially to the inhibition of lipolysis, we feel it would be beyond the scope of this study to select or focus on specific kinases (phosphosites) and generate phosphosite-specific kinase mutant adipocytes to study their individual contributions.

d) Can the authors speculate at least as to how vaspin binding to Lrp1 and internalization go on to modulate PDE activity?

Vaspin inhibition of adrenergic signaling required binding of the LRP1, was independent of intracellular inhibitory G protein signaling (Gai) but suppressed by inhibition of brown adipocyte PDEs using IBMX. Highest expression in AT has been demonstrated for PDE3 and PDE4 and specifically in BAT, PDE4 has been shown to primarily control adrenergic stimulation of lipolysis, while PDE3 seems to predominantly regulate the induction of Ucp1 expression. In line, vaspin inhibition of adrenergically induced lipolysis was dependent on PDE4 activity and not PDE3 (Reviewer Figure 7).

Reviewer Figure 7: Q-R) Inhibition of PDE activities using IBMX: Q) Western blot analysis and R) quantification of basal and CL-induced PKA-activation in differentiated primary brown adipocytes from *VasTg* and WT mice. S-U) Inhibition of PDE3 and PDE4 using *Cilo* and *Ro*: S) Western blot analysis and R) quantification of basal and CL-induced PKA-activation in differentiated primary brown adipocytes from *VasTg* and WT mice with *Cilo* or *Ro* pretreatment. U) Inhibition of total PDE activities using IBMX, *Cilo* or *Ro*: Quantification of basal and CL-induced free fatty acid release with or without vaspin treatment in differentiated imBA ($n = 10-20$ per condition). Data are presented as mean \pm SEM of at least two (Q-U) independent experiments. Statistical significance was evaluated by two-way ANOVA with uncorrected Fischer's LSD (R, T-U). * p -value < 0.05 , ** p -value < 0.01 , *** p -value < 0.001 .

Notably, the abundance of PDE4b and PDE4-anchoring protein myomegalin (PDE4DIP) was significantly higher in vaspin-treated brown adipocytes (Supplementary table 1). We hypothesize that vaspin, via the LRP1, may affect cAMP levels to substantially dampen the BAT thermogenic response. The interaction of LRP1 with MAP kinases ERK1 and 2 may provide a link to the regulation of PDE4, as e.g. ERK2-mediated phosphorylation of PDE isoenzymes PDE4B and PDE4D can have activating and inhibitory effects on PDE4 activity. Related to this, lactoferrin has been shown to induce white adipocyte lipolysis by binding to the LRP1 and activating ERK signaling to increase cAMP levels either involving or not involving Gas signaling, depending on the cell type investigated. The dynamics of LRP1 distribution in the cell-membrane, whether localizing within lipid-rafts or lipid raft-free membrane compartments is believed to be key for the receptors numerous, ligand and tissue/cell-specific signaling (lipid rafts) or endocytotic (raft-free areas) activities and their regulation. Interestingly, inhibition of clathrin-dependent endocytosis abrogated the inhibitory effect of vaspin on lipolysis. This suggests that co-internalization of potentially entire membrane protein complexes or intracellular proteins scaffolds interacting with the LRP1 may be co-internalized may be required for the vaspin effect in thermogenic adipocytes. These protein complexes co-internalized with the respective scaffold receptor remain as active signaling complexes on the surface of endosomal vesicles, essential for the regulation of localized signaling as shown for MAPK/ERK signaling. Clearly, further research is warranted to dissect these distinct ligand- and cell-specific signaling events initiated by LRP1 binding.

We have incorporated this in the revised Figure 3 and the revised discussion.

Minor Points:

1. In Figure 1D and 1P, glucose levels increase after cold exposure in the fasted state. This is somewhat surprising as prior studies have shown that cold exposure in fasted mice leads to an acute decrease in glucose. Can the authors clarify?

We agree with the reviewer and we did not expect to observe elevated blood glucose levels independent of fasting. In the literature, acute cold exposure generally resulted in decreased blood glucose in fed healthy animals (PMID: 25681456, PMID: 26772600). However, in fasted subjects, both humans and animals, studies have shown either no effect of acute cold exposure on blood glucose levels (PMID: 24949663, PMID: 37802078, PMID: 6259583, PMID: 2876821) or an increase in blood glucose (PMID: 28988822 or PMID: 247536).

Our observation of increased blood glucose after acute cold exposure thus have been reported before. An obvious confounding factor could be related to stress, and chronic cold stress has been shown to increase blood glucose (PMID: 25681456). Thus, it could be stress or strain related, with potentially hepatic glucose output fueling BAT thermogenesis. However, we believe that this does not affect the conclusions of our study.

2. In figure 2, the phosphoproteomics suggests showed increased phosphorylation of PKA substrates (lines 144-147), yet the Western blot in Figure 2 showed decreased phosphorylated PKA substrates. Can the authors explain?

We apologize for not being clear enough on these findings. Indeed, phosphoproteome analyses revealed increased phosphorylation of a specific PKA serine residue (Ser83), but this phosphorylation site has been shown to be of inhibitory nature for PKA activity. Thus, vaspin-induced phosphorylation of PKA at Ser83 is in line with reduced PKA activity in VasTG-derived or vaspin-treated adipocytes (Figure 2).

Reviewer #2 (Remarks to the Author):

This paper reports that the adipokine Vaspin can reduce catecholamine-stimulated adipocyte lipolysis, which is associated with reduced brown fat activity. Vaspin- overexpressing mice exhibit reduced brown fat activity brown fat activity and cold-intolerance associated with diminished activation of PKA-signaling. These effects are recapitulated by acute treatment of mice with recombinant Vaspin. Studies in isolated cells show that Vaspin acts directly on adipocytes via the LRP1 receptor to repress b-adrenergic-induced activation of PKA signaling and lipolysis. It is clear that high levels of Vaspin can repress lipolysis. However, there are some important issues and questions, especially related to the physiological role and significance of this pathway that should be addressed for publication in Nat Commun.

We thank this reviewer for his positive comments on our work and have addressed his concerns as follows.

Major concerns:

1. The studies rely exclusively on overexpression/gain-of-function methods with super-physiological levels of Vaspin. The Vaspin injections produced “four order of magnitude higher than physiological levels”. Cell culture studies also utilize a high dose of Vaspin, and produced modest effects. Further studies utilizing lower and more physiological doses of Vaspin should be presented. Additionally, the central conclusions could be strengthened by loss-of-function models, ie. inhibiting receptor in BAT, reducing Vaspin levels/activity?

It is true that the VasTG mouse model exhibits significantly higher vaspin expression levels. However, in humans, vaspin serum levels range from 0.5-1.5 ng/ml (doi: 10.1016/j.diabres.2014.07.026), with some individuals reaching >30 ng/ml (doi: 10.1210/jc.2011-3297), and they show substantial diurnal variation (doi: 10.1210/jc.2009-1088), peaking at 250% of the daily minimum. While the VasTG model has supraphysiologic vaspin levels, they remain within a range observed in human subpopulations and used in in vitro studies, suggesting that this effect still holds physiological relevance.

We have also repeated ELISA measurements of vaspin levels in mice injected with recombinant vaspin (n=8) using a new ELISA kit (**Reviewer Figure 8**). These results confirm the significant increase in vaspin levels followed by a decrease up to 120 min post injection. However, using optimized dilutions for vaspin measurements, the increase in circulating vaspin was now measured to reach 300-400 ng/ml (15-30 min post injection), decreasing to 40 ng/ml at 120

min post injection. These levels are 2-3 magnitudes of order higher than physiological and correlate with our does used for in vitro studies.

Reviewer Figure 8: Levels of human vaspin/SERPINA12 in blood of C57BL/6N mice at indicated times after vaspin injection (i.p., 1 mg/kg, n = 8, except 15 min with n=4); shown as box with whiskers and min/max. Statistical significance was evaluated by two-way ANOVA with Dunnett's (M) post hoc test. *p-value < 0.05, **p-value < 0.01, ***p-value < 0.001. **This panel is shown as Supplementary Figure 1M in the revised manuscript.**

Furthermore, as also mentioned above in response to reviewer one, we fully agree and have in parallel generated BAT-specific (Ucp1-Cre driven) LRP1-BAT-KO mice. We only have preliminary data, but these are consistent with a regulatory function of LRP1 in brown adipocytes (**Reviewer Figures 1-3**).

First, we find that primary brown adipocytes of LRP1-BAT-KO show full differentiation capacity and similar lipid incorporation as wild-type cells (**Reviewer Figure 1A-B**). While in vitro Lrp1 knockdown was significant, but only roughly 25% (**Reviewer Figure 1C**), thermogenic genes such as Ucp1, Adrb3, and Cidea were significantly higher expressed in LRP1-BAT-KO adipocytes (**Reviewer Figure 1D**). Focusing regulation of lipolysis, we observed similar, rather lower FSK-induced PKA activation (**Reviewer Figure 1E-F**), however, Seahorse assays revealed significantly higher respiratory response to FSK and higher maximal respiration in fully differentiated primary brown adipocytes of LRP1-BAT-KO mice (**Reviewer Figure 1G-H**). We observed the same results when stimulating adipocytes with NE (not shown). These results are consistent with our observation of higher lipolysis in CL-treated imBA cells after siLrp1 knock-down (**Figure 3Q**).

Notably, previously generated adipocyte-specific LRP1-AT-KO mice (using the Fabp4/Ap2-Cre) had significantly lower body weight and fat mass due to impaired lipid incorporation and adipocyte dysfunction. Male and female LRP1-BAT-KO mice only show a trend for lower body weights and otherwise have normal body length, body temperature and in trend lower blood glucose levels (**Reviewer Figure 2A-D, F-I**) and importantly, they show no differences in AT mass (eWAT, iWAT and BAT; **Reviewer Figure 2E, J**). Consistent with a regulatory role of LRP1 in mature brown or thermogenic adipocytes, first data on energy expenditure at room temperature in mice, selected for similar lean and fat mass (**Reviewer Figure 3A-B**), showed significantly higher energy expenditure during the active (dark) phase (**Reviewer Figure 3C-D**).

These preliminary data are in line with a regulatory and likely suppressive function of the LRP1 in mature brown or thermogenic adipocytes.

However, as these are preliminary data and ongoing studies, we would prefer to not include them in this manuscript but wanted to share them with the reviewers.

2. Related to #1, It would be important to evaluate the levels of Vaspin (circulating and

adipose tissue) in response to cold exposure, fasting, etc. It is unclear whether this pathway is regulated by physiological (or pathological) stimuli.

We have previously reported regulation of vaspin expression in BAT and both WAT depots to be specifically induced in BAT after cold-exposure, with higher expression also in iWAT. (Weiner et al. Mol Metab 2017). Increased vaspin expression in AT after cold-exposure was not reflected in serum levels though, as these were significantly lower after cold-exposure. This however, as also discussed in our manuscript (lines 277-279), this is consistent with an auto/paracrine mechanism of action by vaspin in thermogenic adipocytes that is based on binding of LRP1, subsequent internalization and resulting clearance from the circulation.

Similarly, vaspin serum levels have been shown to decrease after acute insulin administration during insulin tolerance tests in humans. This may also be explained by insulin-stimulated increase of cell surface LRP1 (as major component of GLUT4 vesicles) and subsequent vaspin binding and internalization, as also shown in insulin-treated adipocytes in our previous work (Tindall et al. FEBSJ 2024).

We have further analyzed expression of endogenous (mouse) and transgenic (human) vaspin in response to an overnight fast in mice (**Reviewer Figure 9A**). In WT C57BL/6N mice, vaspin expression was only affected in iWAT after fasting with a significant reduction. Reduced vaspin expression in the iWAT depot, which is prone to browning, may reflect physiological response supporting BAT heat production under nutritionally challenging conditions requiring increased WAT lipolytic activity and lipid release/supply.

We also measured transgenic vaspin in serum at various ambient temperatures and after cold exposure (**Reviewer Figure 9B**). Also here, vaspin levels at mild cold (23°C) and after acute cold exposure (6h) were lower than in mice with long term adjusted high (cold-exposed) or low/off (thermoneutral) BAT activity, whether activity is high after chronic cold exposure or off under thermoneutral conditions. These observations also are consistent with a mechanism involving receptor-binding, internalization of vaspin in thermogenic adipocytes and clearance from the circulation.

Reviewer Figure 9: A) *SerpinA12* gene expression in eWAT, iWAT and BAT on fed (black circles) and fasted (white circles) wild-type C57BL/6N mice. B) Serum levels of human vaspin in VasTG mice held at thermoneutrality (30°C), mild cold stress (room temperature) or in the cold (acutely and longterm). Data are shown as mean \pm SEM. Statistical significance was evaluated by One way ANOVA with Tukey's or Dunnett's posttest; * $p < 0.05$; ** $p < 0.01$, *** $p < 0.001$. **B) is also shown in Figure 1E**

3. The effect of Vaspin on mitochondrial function/cell respiration is not sufficiently explained. Fig. 2K, L show that basal respiration is suppressed by Vaspin, prior to FSK treatment. Indeed, the magnitude of this reduction (ie. fold reduction) appears similar to that observed after FSK treatment. This is presumably independent of lipolysis and fatty acid oxidation? This effect may contribute greatly to the in vivo phenotypes and should be assessed further.

We have also investigated reasons for initially reduced basal respiration rates, but analysis of mtDNA content or OXPHOS protein expression did not reveal significant differences between genotypes (Supplementary Figure 1G-H).

To assess the suppression of the OCR independent of basal OCR differences, we normalized / baseline-corrected OCR curves for both genotypes to basal values, resulting in individually normalized OCR curves for the MST experiment (**Reviewer Figure 10A-B**). Also, after adjusting for different basal OCR rates, the blunted response to acute adrenergic stimulation remains obvious and significant. The non-significantly higher proton leak and maximal respiratory capacity again argues for acute inhibition of lipolysis-dependent fueling independent from the adipocytes' intrinsic respiratory and uncoupling capacity.

Reviewer Figure 10: A) Time-resolved Oxygen Consumption Rate of differentiated primary brown adipocytes from VasTg and WT mice measured by Seahorse (representative experiment, $n = 5/5$) and B) Baseline-corrected (basal respiration) OCR of the same Seahorse experiment. It remains clear that VasTG primary brown adipocytes show a blunted response to FSK while having normal maximal respiratory capacity. **A) is also shown in Figure 2K**

Minor

4. Fig 4J. This analysis is not clearly described. Are these genes/pathways that correlate with higher vs. lower vaspin expression in human fat tissue? How was this done?

The gene set enrichment analysis (GSEA) was done by first identifying all genes showing a significant (adjusted P) correlation with vaspin (SERPINA12) expression, and subsequently and separately investigating enriched pathways in positively and negatively correlating genes.

Methods section:

To predict the gene function of SERPINA12 and identify potential associations with Kyoto Encyclopedia of Genes and Genomes (KEGG) pathways [61], we conducted Gene Set Enrichment Analysis (GSEA, [62]) of the RNA sequencing data from the LOBB cross-sectional cohort using CorrelationAnalyzer (v1.0.0; [63]). The single gene mode R package was employed, considering the co-expression correlations of SERPINA12 with other genes in the dataset. Genome-wide Pearson correlations were employed as a pre-ranking metric for the GSEA algorithm. The correction for multiple testing was performed using the false discovery rate (FDR) method.

5. It should be clearly stated that the transgene is driven from the aP2/Fabp4-promoter, which is known to have expression in many tissues, including in the brain from early development.

The point has also been made by reviewer one. We have now repeated analysis of transgene expression (human vaspin / SERPINA12) in a tissue panel of VasTG mice showing exclusive expression in AT depots (Reviewer Figure 1A, see above). We did not observe transgene expression in brain, like Prdm16Tg Prdm16xFabp4 mice (PMID: 29131158). We also

fractionated adipose tissue of VasTG mice into mature adipocytes and stromal vascular fraction (SVF) and quantified human vaspin mRNA expression in each fraction. Our data confirm that vaspin is specifically expressed in adipocytes (Reviewer Figure 1B), minimizing concerns about non-adipocyte contributions. Consistently, in we observed the inhibitory vaspin effect on lipolysis in our in vitro experiments using isolated primary SVF-derived brown and white adipocytes from VasTG mice showing inhibited lipolytic response after adrenergic stimulation. We explicitly acknowledge the limitations of the aP2 promoter in the Discussion.

Reviewer #3 (Remarks to the Author):

The present manuscript by Rapöhn et al. reported that vaspin or SEPINA12, an adipocyte-derived serine protease inhibitor, represses BAT thermogenesis by attenuating adrenergic signaling and lipolysis in brown adipocytes. The authors proposed a model in which vaspin binds to the LRP1 receptor on the cell surface and stimulates PDEs, resulting in reduced cAMP-PKA signaling and impaired brown fat thermogenesis. However, several major issues are identified.

1. A previous study by the authors' group reported that transgenic mice expressing vaspin in adipose tissues, driven by the aP2 promoter, limited high-fat diet-induced obesity, improved glycemic control, and elevated whole-body energy expenditure (PMID: 37152954). In contrast, the present work showed that vaspin overexpression attenuates brown fat thermogenesis. It is unclear how these apparently contradictory results can be reconciled because impaired lipolysis would lead to reduced fatty acid oxidation in brown fat, skeletal muscle, and other organs. If a compensatory mechanism exists, the authors need to address such mechanisms.

This is an important point, and we agree that this should be more explicitly discussed.

The seemingly contradictory effects of vaspin on energy balance can be reconciled by considering its distinct direct and indirect actions. In the present study, we demonstrate that vaspin directly inhibits adrenergic activation of lipolysis, thereby restricting fatty acid supply for thermogenesis in brown adipose tissue (BAT) under acute metabolic demands such as cold exposure or fasting. This explains the observed attenuation of thermogenesis in BAT. However, in the context of chronic high-fat diet (HFD) feeding, vaspin's predominant effect is its indirect anti-inflammatory action, which preserves thermogenic adipose tissue function by preventing adipose tissue inflammation and dysfunction. This protective effect likely outweighs the acute inhibitory effect on lipolysis, leading to improved whole-body energy expenditure and metabolic health.

Additionally, compensatory mechanisms may contribute to maintaining energy homeostasis in vaspin-overexpressing models. For instance, improved insulin sensitivity and enhanced glucose utilization in BAT and skeletal muscle may provide alternative substrates for thermogenesis and energy expenditure under HFD conditions. Further investigation is needed to delineate these compensatory pathways and their relative contributions to energy metabolism in different physiological and dietary states.

We have discussed this in the revised manuscript.

2. The authors used the aP2 promoter to drive vaspin in their transgenic mice. It is now well recognized that the aP2 promoter is expressed in many tissues other than adipocytes, and as

such, it is unclear how vaspin acts on adipocyte lipolysis as a secretory molecule or acts intracellularly. Even though the authors performed cultured cell experiments, the dose used in the study (0.5 ug/ml) was far higher than a physiological dose.

We analyzed transgene expression (human vaspin / SERPINA12) in a tissue panel of VasTG mice as well as in iWAT fractions of mature adipocytes and SVF from WT and VasTG mice (n=3-4 per genotype, see **Reviewer Figure 1**). The results clearly show that transgene expression is specific for mature adipocytes. We did not observe transgene expression in brain, similar to (Prdm16Tg Prdm16xFabp4 mice, PMID: 29131158).

As mentioned above, it is true that the VasTG mouse model exhibits significantly higher vaspin expression levels. However, in humans, vaspin serum levels range from 0.5-1.5 ng/ml (doi: 10.1016/j.diabetes.2014.07.026), with some individuals reaching >30 ng/ml (doi: 10.1210/jc.2011-3297), and they show substantial diurnal variation (doi: 10.1210/jc.2009-1088), peaking at 250% of the daily minimum. While the VasTG model has supraphysiologic vaspin levels, they remain within a range observed in human subpopulations and used in in vitro studies, suggesting that this effect still holds physiological relevance.

3. Related to the above point, the efficacy of PKA blots appeared potent when transgenic mice-derived adipocytes were used compared to ones with purified vaspin (e.g., Fig2D, E). It is conceivable that vaspin acts intercellularly rather than the LRP1-mediated signaling pathway.

This related to a comment from reviewer one. As also mentioned above, we thank the reviewer for these suggestions and have performed additional experiments to address whether vaspin needs to be secreted and acts extracellularly (using BFA) and whether the inhibitory effect on lipolysis requires cell surface LRP1 (using RAP pretreatment) and is dependent on internalization (using chlorpromazine, cpz) (**Reviewer Figure 6**, above).

We found that blocking of protein secretion using BFA abolished the inhibitory effect in primary brown adipocytes from VasTG mice. We then pretreated imBA with RAP, to induce ligand-bound internalization and lower LRP1 receptor density on the cell surface before analyzing the effect of vaspin on CL-induced lipolysis. As already shown, RAP treatment (and interaction with LRP1) alone did not affect PKA-activation and lipolysis but did prevent vaspin-mediated inhibition of lipolysis. This was also observed for inhibition of clathrin-mediated internalization, and cpz prevented vaspin inhibition of CL-induced lipolysis.

Along these lines, a second mechanism may contribute to the more pronounced effect of vaspin observed when using VasTG primary adipocytes compared to recombinant vaspin in wildtype cells. In addition to neutral lipolysis, thermogenic adipocytes also rely on lysosomal lipolysis mediated by lysosomal acidic lipase (LAL) from autophagocytosed lipid droplets or endocytosed lipoproteins (Duta-Mare et al. 2018 BBA - Mol Cell Biol Lipids). Endocytosis of lipoproteins is facilitated by members of the LDL receptor family (mainly LDLR, vLDLR and LRP1, Espirito Santo et al. 2005 J Lipid Res) in lipid rafts or by clathrin-dependent internalization. Vaspin's ability to influence lysosomal lipid trafficking through LDLR family interactions presents a potential mechanism for controlling lipid turnover. And continuous inhibition of this pillar of thermogenic adipocyte fuel supply in VasTG primary adipocytes may increase the observed antilipolytic effect compared to brown adipocytes acutely treated with recombinant vaspin. To test this hypothesis, we utilized ELISA-based binding assays to screen for vaspin receptors within the LDL receptor family. We obtained recombinant receptor ligand binding domains of the LDLR, vLDLR, LRP2, LRP4, LRP5, LRP6 and LRP10. Vaspin did not bind LRPs 5, 6

and 10. However, in addition to previously identified LRP1, vaspin exhibited low, in part sub-nanomolar affinities for LDLR and vLDLR, as well as LRP2 and LRP4 (Table 1 in the revised manuscript) (**Reviewer Figure 11**).

Reviewer Figure 11: V-X) ELISA-based analysis of TAMRA-vaspin binding to LDL receptors (V; LDLR, vLDLR), LRP receptors (W; LRP2, LRP4-6, LRP10) and class B scavenger receptors (X; SR-BI, CD36). Nonlinear regression analysis was performed to determine EC50 presented in Table 1.

Previous studies have excluded LDLR and LRP1 as significant contributors to triglyceride-rich lipoprotein uptake in BAT using whole body knock-out (LDLR) or overexpressing (LRP1) mice (Bartelt et al. 2011, Nat Med). However, from all potential endocytosis receptors vaspin most strongly bound the VLDLR. The VLDLR is significantly higher expressed in thermogenic (brown and beige) adipocytes compared to white (Perdikari et al. 2018 Cell Rep) and VLDLR-mediated VLDL uptake is a crucial pathway fueling thermogenesis in BAT (Shin et al. 2022 Cell Rep). Thus, binding to lipoprotein endocytosis receptors of the LDL receptor family together with adaptive reductions in thermogenic gene expression likely enhanced the antilipolytic effect in VasTg expressing cells compared to acutely vaspin-treated wildtype adipocytes. As this is the first report of vaspin binding to different members of the LDL receptor family, there are many new avenues of research to be pursued to investigate the cellular consequences of individual receptor binding events, that will not only affect adipocytes, but liver and vascular lipid uptake as well.

Taken together, these data suggest that vaspin exerts multiple mechanisms to control adrenergic activation of brown adipocyte metabolism and lipid turnover through interactions with the LDLR family, thereby inhibiting cAMP-dependent neutral lipolysis via modulation of PDE activities and blocking endocytotic lipid trafficking towards lysosomal acid lipolysis.

We have included this data in the revised manuscript in Figure 3, Supplementary Figure 4.

4. For the PKA data in Fig.3D, G, J, Fig.4F, and 4M, the effect sizes were low, raising a concern about the robustness of the experiments.

We agree and have tried to find replacement for the pPKA substrate antibody. The original antibody from Cell Signalling Technologies (a very good, frequently used and referenced antibody). However, the quality of Western blot data (sensitivity, background, band pattern) has changed and now is substantially worse and different. We also tried PKA activity assays from Abcam, but did not obtain robust results under control conditions (wild type adipocytes +/- adrenergic stimulation) and also found no references in the literature demonstrating the usefulness of these assays (at least in adipocytes).

We are aware that this is not ideal, but unfortunately there is no alternative to the best of our knowledge. Therefore, we used lipolysis assays in addition, as these were more robust.

5. The authors proposed a model in which vaspin acts on PDEs by using IBMX, but the underlying mechanism was not explored. It is possible that vaspin and PDEs are completely parallel pathways.

Vaspin inhibition of adrenergic signaling required binding of the LRP1, was independent of intracellular inhibitory G protein signaling (Gai) but suppressed by inhibition of brown adipocyte PDEs using IBMX. Highest expression in AT has been demonstrated for PDE3 and PDE4 and in specifically in BAT, PDE4 has been shown to primarily control adrenergic stimulation of lipolysis, while PDE3 seems to predominantly regulate the induction of Ucp1 expression. In line, vaspin inhibition of adrenergically induced lipolysis was dependent on PDE4 activity and not PDE3 (**Reviewer Figure 7**, above). Notably, the abundance of PDE4b and PDE4-anchoring protein myomegalin (PDE4DIP) was significantly higher in vaspin-treated brown adipocytes (Supplementary table 1). We hypothesize that vaspin, via the LRP1, may affect cAMP levels to substantially dampen the BAT thermogenic response. The interaction of LRP1 with MAP kinases ERK1 and 2 may provide a link to the regulation of PDE4, as e.g. ERK2-mediated phosphorylation of PDE isoenzymes PDE4B and PDE4D can have activating and inhibitory effects on PDE4 activity. Related to this, lactoferrin has been shown to induce white adipocyte lipolysis by binding to the LRP1 and activating ERK signaling to increase cAMP levels either involving or not involving Gas signaling, depending on the cell type investigated. The dynamics of LRP1 distribution in the cell-membrane, whether localizing within lipid-rafts or lipid raft-free membrane compartments is believed to be key for the receptors numerous, ligand and tissue/cell-specific signaling (lipid rafts) or endocytotic (raft-free areas) activities and their regulation. Interestingly, inhibition of clathrin-dependent endocytosis abrogated the inhibitory effect of vaspin on lipolysis (**Reviewer Figure 6**, above). This suggests that co-internalization of potentially entire membrane protein complexes or intracellular proteins scaffolds interacting with the LRP1 may be co-internalized may be required for the vaspin effect in thermogenic adipocytes. These protein complexes co-internalized with the respective scaffold receptor remain as active signaling complexes on the surface of endosomal vesicles, essential for the regulation of localized signaling as shown for MAPK/ERK signaling. Clearly, further research is warranted to dissect these distinct ligand- and cell-specific signaling events initiated by LRP1 binding.

We have incorporated this in the revised discussion.

We appreciate and thank all reviewers very much for their critical assessment of our work and for their constructive and helpful comments that have very much improved this manuscript.

Reviewer #1 (Remarks to the Author):

The revised manuscript has been substantially improved with the addition of new experimental data. The combination of these data and those shown in the rebuttal have addressed all of the concerns raised in my initial review. This is a nicely done study that will be of interest to people in the field.

We thank this reviewer.

Reviewer #2 (Remarks to the Author):

The authors have addressed many of the reviewers' previous critiques. The paper is well presented and the conclusions are generally well supported. There are a couple of remaining points that should be addressed before publication.

We thank this reviewer and have addressed these points as outlined below.

1. I am still unclear about the seemingly striking suppressive effect of Vaspin OE on the basal respiration of brown adipocytes isolated from mice (Figure 2K). The proteomic analyses in the paper indicates that respiratory proteins were reduced in Vaspin-treated brown adipocytes. From Results: "Among the proteins with lower abundance, pathway analysis revealed significant enrichment of proteins related to oxidative phosphorylation, such as core subunits of respiratory chain complex I (NDUFS6, NDUFA3, NDUFA12 and NDUF8), complex III (UQCRC1), and COX7A2, the terminal oxidase in mitochondrial electron transport (KEGG mouse 2019, Figure 2C and Supplementary Table 7)." However, there appears to be no effect on these components in vivo (Fig. S1), and as alluded to in the reviewer response - can the authors add to the discussion of this point. Is this also LRP dependent?

Indeed, when analyzing whole BAT tissue in VasTg mice (SF1 G-H), we did not observe significant changes in OXPHOS protein expression. However, it must be acknowledged that instead of isolated adipocytes (as in the proteomic analyses) we here have bulk tissue and other cell types in BAT may mask changes occurring specifically in adipocytes. Furthermore, while using the OXPHOS antibody cocktail is a robust and convenient way to assess principal differences in key mitochondrial proteins, it is not ideally suited to confirm our proteomics data that revealed significant differences in more specific OXPHOS proteins and subunits that are not covered by the antibody cocktail. We have addressed these limitations in the revised manuscript.

Whether these effects are mediated by LRP1 remains not fully understood. We have now analyzed mitochondrial protein expression (using the OXPHOS ab cocktail) in cold-exposed BAT-specific Lrp1KO mice and found significantly higher abundance of complex II (SDHB) without changes in the other complexes detected by the antibody cocktail. While this may suggest, in line with the inhibitory action of the vaspin-LRP1 axis, that

knock-down of LRP1 in BAT may rather induce OXPHOS protein expression, full proteomic analyses are required to assess this in more detail.

Reviewer Figure 1. OXPHOS protein expression in male BAT-specific *Lrp1* knockout mice. (A) Western blot analysis of LRP1 (A, with quantification below) and OXPHOS complexes (B, quantification in C) in BAT of Protein expression is normalized to WT littermate controls. Data are shown as mean \pm SEM. Statistical significance was calculated by t-test (A) or two-way ANOVA with Šídák's post-hoc test (C). *p-value < 0.05.

2. Fig 3B-F. This experiment is not very informative without having the +CL and no BFA condition. The reader can't appreciate the effect of secretion without seeing the untreated (no BFA) condition in the same experiment.

We repeated the experiment adding wells for the +CL – BFA conditions which gave the same result. Notably, total NEFA release is blunted with BFA in both WT and VasTg primary brown adipocytes, however there is no additional inhibitory effect in VasTg adipocytes. We have included these data in the revised manuscript.

Reviewer Figure 2. Basal and CL-induced free fatty acid (NEFA) release in differentiated primary brown adipocytes from VasTg (orange) and WT (white) mice after blocking vaspin secretion (using BFA; n = 5 - 12 per condition). Data are shown as mean \pm SEM. Statistical significance was calculated by one-way ANOVA with Šídák's post-hoc test. ***p-value < 0.001.

This figure has been included in the revised manuscript as Figure 3E.

3. Determining the effect of Vaspin on FFA levels in vivo would strengthen the conclusions. For example, is there a decrease in FFA levels in vivo in response to CL administration? Or during cold exposure/fasting?

We agree and have checked for available serum samples.

In the literature, acute (short term 1-6h) cold exposure mostly had no significant effect on circulating FFA levels in rodents and humans (Vallerand and Jacobs 1990, Gagnon et al. 2013, Weng et al. 2023), however significant increases in circulating FFA levels were observed after acute cold exposure (2h-6h) in mice and humans (Imai et al. 2006, Straat et al. 2022).

A recent study reported (Bornstein et al. 2023) extensive flux studies during cold exposure in both fed and fasted mice. They found that in fasted mice, cold exposure drives a large increase in circulating FFA flux, fueling most of the rise in systemic CO₂ production. In fed mice, cold stress only modestly raised fatty acid flux, but fatty acids still likely served as the main oxidative fuel in BAT through provision of acetyl-CoA.

We could measure FFA in serum of acutely cold exposed (and fasted) WT and VasTg mice from the 4 h time point (manuscript Figure 1C, Reviewer Figure 2). FFA levels in WT littermates were significantly higher than in VasTg mice, consistent with limited activation of WAT lipolysis to fuel BAT thermogenesis. We unfortunately do not have serum from these animals before start of the experiment.

Reviewer Figure 3. F1C) Rectal body temperature during acute cold exposure in fasted (C, n = 8/7) VasTg and WT mice. RF2) Blood NEFA levels of VasTg and WT mice after acute (4 h) cold exposure (n = 4/4). Data are shown as mean ± SEM. Statistical significance was calculated by two-way ANOVA with Šídák's (F1C) or unpaired t-test (RF2). *p-value < 0.05, ***p-value < 0.001.

These new data have been included in the revised manuscript as Figure 1E.

Reviewer #3 (Remarks to the Author):

The authors provided new data that addresses the reviewer's comments satisfactorily. Congratulations!

We thank this reviewer.

References:

- Bornstein, M. R., M. D. Neinast, X. Zeng, Q. Chu, J. Axsom, C. Thorsheim, K. Li, M. C. Blair, J. D. Rabinowitz and Z. Arany (2023). "Comprehensive quantification of metabolic flux during acute cold stress in mice." *Cell Metab* **35**(11): 2077-2092 e2076.
- Gagnon, D. D., H. Rintamaki, S. S. Gagnon, S. S. Cheung, K. H. Herzig, K. Porvari and H. Kyrolainen (2013). "Cold exposure enhances fat utilization but not non-esterified fatty acids, glycerol or catecholamines availability during submaximal walking and running." *Front Physiol* **4**: 99.

Imai, J., H. Katagiri, T. Yamada, Y. Ishigaki, T. Ogihara, K. Uno, Y. Hasegawa, J. Gao, H. Ishihara, H. Sasano and Y. Oka (2006). "Cold exposure suppresses serum adiponectin levels through sympathetic nerve activation in mice." Obesity (Silver Spring) **14**(7): 1132-1141.

Straat, M. E., L. Jurado-Fasoli, Z. Ying, K. J. Nahon, L. G. M. Janssen, M. R. Boon, G. F. Grabner, S. Kooijman, R. Zimmermann, M. Giera, P. C. N. Rensen and B. Martinez-Tellez (2022). "Cold exposure induces dynamic changes in circulating triacylglycerol species, which is dependent on intracellular lipolysis: A randomized cross-over trial." EBioMedicine **86**: 104349.

Vallerand, A. L. and I. Jacobs (1990). "Influence of cold exposure on plasma triglyceride clearance in humans." Metabolism **39**(11): 1211-1218.

Weng, X., C. Wang, Y. U. Yuan, Z. Wang, J. Kuang, X. U. Yan and H. Chen (2023). "Effect of Cold Exposure and Exercise on Insulin Sensitivity and Serum Free Fatty Acids in Obese Rats." Med Sci Sports Exerc **55**(8): 1409-1415.